# Lipoprotein metabolism mediates hematopoietic stem cell responses under acute anemic conditions

Kiyoka Saito[1,2,10], Mark van der Garde[2,3,10], Terumasa Umemoto [1], Natsumi Miharada [1], Julia Sjöberg [2], Valgardur Sigurdsson[2], Haruki Shirozu[1], Shunsuke Kamei[4], Visnja Radulovic[2], Mitsuyoshi Suzuki [2,5], Satoshi Nakano[2,5], Stefan Lang [6], Jenny Hansson [7], Martin L. Olsson [8], Takashi Minami [4], Gunnar Gouras [9], Johan Flygare [2] & Kenichi Miharada [1,2] ✉

Hematopoietic stem cells (HSCs) react to various stress conditions. However, it is unclear whether and how HSCs respond to severe anemia. Here, we demonstrate that upon induction of acute anemia, HSCs rapidly proliferate and enhance their erythroid differentiation potential. In severe anemia, lipoprotein profiles largely change and the concentration of ApoE increases. In HSCs, transcription levels of lipid metabolism-related genes, such as very low-density lipoprotein receptor (*Vldlr*), are upregulated. Stimulation of HSCs with ApoE enhances their erythroid potential, whereas HSCs in *Apoe* knockout mice do not respond to anemia induction. Vldlr^high^HSCs show higher erythroid potential, which is enhanced after acute anemia induction. Vldlr^high^HSCs are epigenetically distinct because of their low chromatin accessibility, and more chromatin regions are closed upon acute anemia induction. Chromatin regions closed upon acute anemia induction are mainly binding sites of Erg. Inhibition of Erg enhanced the erythroid differentiation potential of HSCs. Our findings indicate that lipoprotein metabolism plays an important role in HSC regulation under severe anemic conditions.

While the majority of adult hematopoietic stem cells (HSCs) are maintained in a dormant state under steady-state conditions[1], various forms of hematopoietic stress, such as myeloablation and infection, promote their proliferation and rapid production of desired downstream progenies[2–5]. In addition, inflammatory cytokines can trigger HSC growth and biased differentiation[6–9]. However, it is unclear whether and how HSCs respond to severe anemia. Red blood cells (RBCs) are the most abundant cell types in the body, representing more than

80% of the somatic cells[10]. Since erythroid progenitors, especially the burst-forming unit erythroid (BFU-E), self-renew under stress erythropoiesis conditions[11,12], it has been assumed that HSCs do not react to severe anemia. In addition, early studies have suggested that erythropoietin (EPO), a major cytokine that promotes erythropoiesis, only targets committed erythroid progenitor cells or downstream progeny based on the expression pattern of its receptor (EPO-R)[13–15]. Nonetheless, recent studies have reported the expression of EPO-R in a

[1]International Research Center for Medical Sciences, Kumamoto University, Kumamoto, Japan. [2]Division of Molecular Medicine and Gene Therapy, Lund Stem Cell Center, Lund University, Lund, Sweden. [3]Department of Medicine III, Hematology and Oncology, Technical University of Munich, Munich, Germany. [4]Division of Molecular and Vascular Biology, Institute of Resource Development and Analysis, Kumamoto University, Kumamoto, Japan. [5]Department of Pediatrics, Juntendo University Faculty of Medicine, Tokyo, Japan. [6]StemTherapy Bioinformatics Core facility, Lund Stem Cell Center, Lund University, Lund, Sweden. [7]Division of Molecular Hematology, Lund Stem Cell Center, Lund University, Lund, Sweden. [8]Division of Hematology and Transfusion Medicine, Lund University, Lund, Sweden. [9]Department of Experimental Medical Sciences, Lund University, Lund, Sweden. [10]These authors contributed equally: Kiyoka Saito, Mark van der Garde. ✉e-mail: kenmiharada@kumamoto-u.ac.jp

broad range of cells, including HSCs[16–18]. Yet, the effect of EPO on cells higher in the hematopoietic hierarchy, particularly under acute anemic stress, is still controversial, as many of these studies utilized artificial EPO-overexpressing systems[17,19]. Other factors contributing to erythropoiesis, such as IGF-1, BMP4, and glucocorticoids, also enhance the proliferation and differentiation of erythroid progenitors and precursors[11,20–22], leaving the mechanisms governing the enhanced erythropoiesis potential of HSCs unclear.

Lipoproteins are complexes of cholesterol esters, triacylglycerols, phospholipids, and apolipoproteins, which play a crucial role in transporting cholesterol and lipids[23]. Depending on their density and size, lipoproteins are usually categorized into four groups: large to small, chylomicron (CM), very low-density lipoprotein (VLDL), low-density lipoprotein (LDL), and high-density lipoprotein (HDL). Nascent VLDL is synthesized in the liver, and by receiving apolipoproteins from HDL, it becomes mature VLDL in the bloodstream. Circulating VLDL is hydrolyzed and releases glycerol, fatty acids, and some apolipoproteins, finally becoming LDL, which is absorbed by the liver[23,24]. Through their roles in lipid metabolism, lipoproteins have been implicated in various biological reactions and diseases, including immune responses, coronary heart disease, and atherosclerosis[25–27]. It has been demonstrated that HSC proliferation is enhanced in patients with atherosclerosis and mouse models, which is considered to accelerate clonal hematopoiesis[28]. In contrast, HDL is known to suppress HSPCs in the atherosclerosis mouse model[29]. However, their exact roles in the HSC regulation and hematopoietic stress responses, as well as the molecular mechanism behind have been unknown.

In this study, we report that HSCs respond to acute anemia by proliferating and enhancing their erythroid differentiation potential. Notably, we did not find obvious signs of the implication of EPO or other known erythroid regulators in HSC responses; instead, we found the involvement of lipoproteins and their metabolism. Our study revealed that not only progenitor cells but also HSCs respond to severe anemic conditions and contribute to erythropoiesis through rapid expansion and a transient fate change. In addition, we propose a lipoprotein-based regulatory mechanism for HSC differentiation toward the erythroid lineage.

## Results

### HSCs expand upon acute anemia induction

To elucidate the response of HSCs to acute anemia, we used phenylhydrazine (PHZ) injection as a model of hemolytic anemia[30,31]. As it has been demonstrated that the cycling status and erythroid potential of HSCs differ between males and female[32], we compared the responses of both male and female mice upon PHZ injection (Fig. 1a). Peripheral blood (PB) analyses showed that erythroid parameters, including RBC count, hematocrit (HCT) content, and hemoglobin (HGB) concentration, decreased rapidly after PHZ injection, reaching the lowest level on day 3 (Fig. 1b and Supplementary Fig. 1a). While HCT levels returned to physiological levels on day 6, the RBC count and HGB/MCH levels indicated that the mice were still recovering from anemic stress. No noticeable differences were found between the male and female mice. Next, we analyzed the bone marrow (BM) of PHZ-injected mice to monitor the frequency of HSC and progenitor populations. In male BM, the number of HSCs (CD150$^+$CD34$^-$c-kit$^+$Sca-I$^+$Lineage$^-$; CD150$^+$CD34$^-$KSL) and megakaryocyte/erythrocyte lineage-restricted progenitors (MEP; CD34$^-$CD16/32$^-$c-kit$^+$Sca-I$^-$Lineage$^-$)[33] increased immediately after PHZ injection, reaching a peak 3 to 4 days after injection, whereas the number of common myeloid progenitors (CMP; CD34$^+$CD16/32$^-$c-kit$^+$Sca-I$^-$Lineage$^-$) and granulocyte/macrophage lineage-restricted progenitors (GMP; CD34$^+$CD16/32$^+$c-kit$^+$Sca-I$^-$Lineage$^-$) did not show any changes (Fig. 1c–e). Cell cycle assays using Ki-67 showed that HSCs in PHZ-treated mice entered the cell cycle on day 1 and returned to a dormant state on day 3 (Supplementary Fig. 1b,

c). In contrast, female BM showed no significant alterations in HSC and progenitor frequencies after PHZ treatment. To clarify whether the migration of HSCs to the spleen[34] could explain the observed differences between male and female mice, we analyzed the frequency of HSCs in the spleen after PHZ injection. Contrary to the BM, we found a persistent increase in HSC counts in the female spleen, while in the male spleen, HSC counts only showed a transient increase (Fig. 1f). While the increase in splenic MEP was similar in male and female mice, we found that Ter119$^+$ erythroid cells were higher in female mice (Supplementary Fig. 1d, e). We wondered whether HSC proliferation was induced by increased estrogen levels[32]; however, the concentration of estrogen in the PB did not increase upon PHZ injection (Supplementary Fig. 1f). These findings indicated that HSCs from both male and female mice proliferated upon anemia induction, albeit at distinct anatomical locations.

### HSCs enhance erythroid potential under anemic stress

To determine whether HSCs change their lineage commitment upon acute anemia induction, we used a modified colony-forming unit (CFU) assay, in which the frequencies of four hematopoietic lineages (granulocytes, macrophages, erythroids, and megakaryocytes) contained in each colony were analyzed using flow cytometry (CFU-FACS). This prevented observer bias and provided quantitative information on the lineage output. CFU-FACS categorized the formed colonies into the granulocyte-erythroid-macrophage-megakaryocyte colony (GEMMk), granulocyte-erythroid-macrophage colony (GEM), granulocyte-macrophage-megakaryocyte colony (GMMk), erythroid-megakaryocyte colony (EMk), granulocyte-macrophage colony (GM), and macrophage colony (M) (Supplementary Fig. 2a). Using this method, we compared the lineage output of colonies formed by HSCs isolated from steady-state mice and mice under anemic stress. CFU-FACS showed that the frequency of EMk colonies tended to be higher for HSCs isolated from PHZ-injected mice (Supplementary Fig. 2b); however, it was unclear whether erythroid potential per colony (individual HSC) was also higher because a huge variation in lineage balance was found even among GEMMk colonies. Therefore, we compared the frequencies of each lineage in all colonies under different conditions. HSCs isolated from mice 3 days post PHZ injection formed significantly more erythroid cells per colony, whereas the percentage of megakaryocytes and myeloid cells per colony did not change (Fig. 2a). This difference was not seen on day 1 (Supplementary Fig. 2c). As both male and female HSCs showed increased erythroid cells in the colonies, we decided to focus on male HSCs for subsequent experiments. As an alternative method to induce acute anemia, we performed phlebotomy, which also achieved anemic conditions in the treated mice (Supplementary Fig. 2d). CFU-FACS revealed that HSCs of phlebotomized mice also had an enhanced erythroid potential (Fig. 2b), suggesting that the higher erythroid content in the colonies was not due to the impact of PHZ. Because HSC expansion was observed in the spleen (Fig. 1f), especially in female mice, we also compared the erythroid potential of spleen-expanded HSCs after PHZ injection. CFU-FACS revealed that HSCs in the spleen also showed enhanced erythroid differentiation; however, the difference was modest or insignificant (Supplementary Fig. 2e).

To further investigate whether HSCs from anemic mice have a higher erythroid differentiation potential, we performed an in vivo transplantation assay using Kusabira Orange (KuO) mice[35,36], allowing us to track the RBC and platelet progeny of the transplanted HSCs. CD150$^+$CD34$^-$KSL cells were isolated from PBS- or PHZ-injected KuO mice and transplanted into lethally irradiated recipient mice (Fig. 2c). Two weeks after transplantation, KuO$^+$ RBCs and platelets were detected in PB. In mice transplanted with HSCs from PHZ-treated KuO mice, we found higher frequencies of KuO$^+$ RBCs than the control HSC-transplanted mice, and the ratio of RBC/platelet in KuO$^+$ cells was also higher in these mice (Fig. 2d). However, these differences were not observed 4 or 12 weeks post-transplantation, suggesting that the

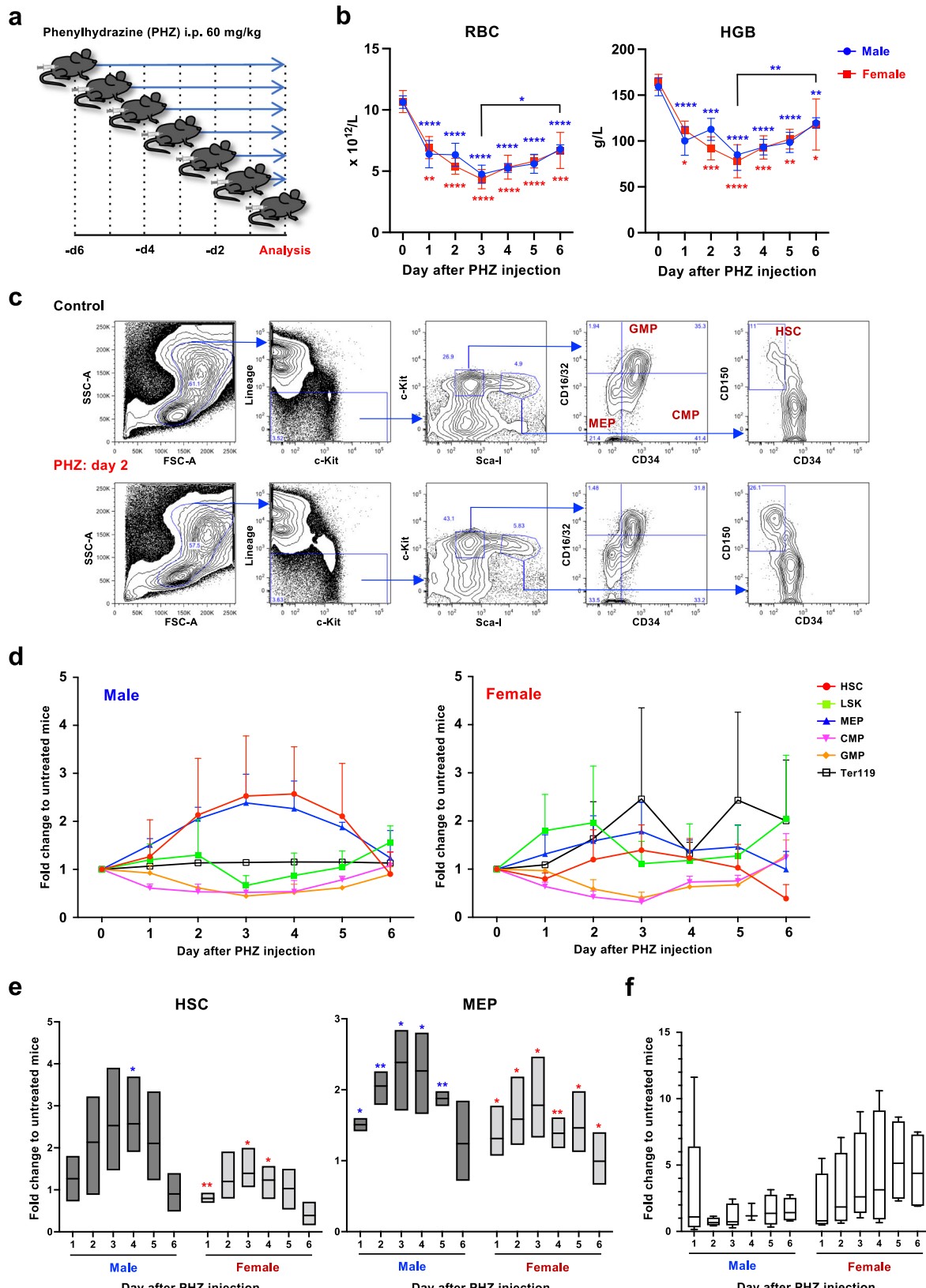

enhanced erythroid potential of HSCs in anemic mice was transient. However, the frequency of KuO⁺ cells within the mononuclear cells did not show a significant difference, indicating that HSCs under acute anemic conditions were still capable of giving rise to multiple lineages (Fig. 2d).

## HSCs under acute anemia elevate erythroid gene expression

We next investigated whether acute anemia induction changes EPO concentration in the BM. We compared EPO concentrations in the blood plasma and BM fluid of PHZ-injected and phlebotomized mice using Enzyme-linked immunosorbent assay (ELISA). As expected, EPO

**Fig. 1 | HSCs expand upon acute anemia induction. a** Experimental design of the PB and BM analysis after hemolysis induction. Hemolytic anemia was induced by intraperitoneally injecting 60 mg/kg of PHZ or the same volume of PBS. The analysis of all groups was done on the same day. The figure was created by authors. **b** PB analysis of the PHZ-treated mice. Red blood cell count (RBC) and hemoglobin concentration (HGB) are shown. See also Supplementary Fig. S1A. Four male and three female mice were used for each time point. Mean ± SEM are shown. Significance was calculated between day 0 (untreated) and each time point unless separately indicated. Adjusted $p$ values were obtained using one-way ANOVA (Tukey's multiple comparison test). **c** Representative FACS plot of hematopoietic stem and progenitor populations in the BM of control (Ctrl) or PHZ-treated mice. Each population was defined as the following: HSC (CD150$^+$CD34$^-$c-kit$^+$Sca-I$^+$Lineage$^-$; CD150$^+$CD34$^-$KSL), granulocyte/macrophage lineage-restricted progenitors (GMP; CD34$^+$CD16/32$^+$c-kit$^+$Sca-I$^-$Lineage$^-$), common myeloid progenitors (CMP; CD34$^+$CD16/32$^-$c-kit$^+$Sca-I$^-$Lineage$^-$), megakaryocyte/erythrocyte lineage-restricted progenitors (MEP; CD34$^-$CD16/32$^-$c-kit$^+$Sca-I$^-$Lineage$^-$). **d** Fold change of the frequencies of HSC and progenitor fractions over experimental time. Fold change compared with day 0 (untreated) mice (mean ± SEM) are shown. $n = 3$. **e** Fold change of HSC (left) and MEP (right) populations in male and female BM. Fold change compared with day 0 (untreated) mice (mean ± SEM) are shown. Six male and three female mice were used for each time point. Each bar extends from the minimum value to the maximum value, with center lines showing the mean value. Four mice for each time point were used. $P$ values were obtained using one sample t and Wilcoxon test. **f** Fold change of HSC in the male and female spleen. Fold change compared with day 0 (untreated) mice (mean ± SEM) are shown. Each bar extends from the minimum value to the maximum value, with center lines showing the mean value. Four mice for each time point were used. Adjusted $P$ values were obtained using one-way ANOVA (Dunnett's multiple comparisons test). *$p < 0.05$, **$p < 0.01$, ***$p < 0.001$, ****$p < 0.0001$. Exact $P$ values are provided as Source Data.

concentration was significantly elevated in the blood plasma of mice with acute anemia (Fig. 3a). However, EPO levels in the BM fluid remained unchanged or only showed a modest increase (Fig. 3b). Concentrations of two major niche factors, SCF and CXCL12[37–40], were higher in the BM fluid than those in the blood plasma but did not increase upon anemia induction (Fig. 3a, b), indicating that proliferation and altered lineage-choice were not due to the effect of EPO or SCF/CXCL12. To uncover the molecular mechanisms governing the acute anemia effect in HSCs, we compared the gene expression profiles of HSCs in mice three days after PHZ injection or phlebotomy (Fig. 3c). Microarray analyses identified 245 significantly upregulated genes and 72 downregulated genes in HSCs isolated from mice three days after PHZ injection, and 132 upregulated genes and 62 downregulated genes after phlebotomy. Among them, 51 and 32 genes were commonly up- or downregulated between PHZ injection and phlebotomy, respectively (Fig. 3d and Supplementary Data 1). These differentially expressed genes did not include typical erythroid-related transcription factors such as *Gata1* and *Eklf* (Fig. 3e and Supplementary Data 1). Although no representative erythroid-related genes were significantly upregulated, gene set enrichment analysis (GSEA)[41] indicated that the gene expression pattern of HSCs under acute anemia was similar to erythroid/megakaryocyte lineage patterns (PRECFU-E, MKP), whereas lymphoid/myeloid signatures were not induced (Fig. 3f and Supplementary Fig. 3a). The inflammation signature was not enriched in acute anemic mice-derived HSCs; in fact, Cytokine-Array did not observe induction of any inflammation cytokines (Supplementary Fig. 3b, c). Instead of known erythropoiesis regulatory mechanisms, commonly upregulated genes were involved in the lipid metabolism, such as fatty acid-binding protein 5 (*Fabp5*) and sterol O-acyltransferase 2 (*Soat2*) (Fig. 3e). GSEA showed that in HSCs of acute anemic mice (both PHZ-treated and phlebotomized), cell cycle and translation-related gene signatures (e.g., E2F target, G2M checkpoint, Myc target, rRNA/tRNA metabolic process, and ribosome biogenesis) were highly enriched (Fig. 3g, h and Supplementary Data 2).

### Reduced scavenging factor response modifies megakaryopoiesis
Commonly downregulated genes included complement-related genes (*C1qa*, *C1qb*, *C1qc*), as well as those involved in heme metabolism and scavenging factors (*Hmox1*, *Cd163*, *Cd68*, and *Lrp1*) (Fig. 3e). Hemolysis induced by PHZ injection releases hemoglobin (Hb) and free-heme into the blood circulation, which is a highly reactive molecule due to its oxygen binding capacity and radical formation; thus, it is toxic to cells and tissues[42]. To capture and detoxify Hb and free heme, many scavenger factors, such as haptoglobin (Hp) and hemopexin (Hpx), play vital roles in removing those factors[43–45]. We detected an increase in Hb concentration in the PB of PHZ-treated animals (Supplementary Fig. 3d), and as expected, Hp concentration in the PB and BM increased 1 day after PHZ injection (Supplementary Fig. 3e) Next, we investigated

whether scavenger factors play a role in controlling the differentiation of HSCs CFU-FACS showed that the addition of recombinant Hp or Hpx to the CFU assay of normal HSCs increased the frequency of megakaryocytes and decreased the number of erythroid cells (Supplementary Fig. 3f).

One of the most significantly downregulated genes upon both PHZ treatment and phlebotomy was *Cd163* (Fig. 3d), which is a receptor for Hb and Hp-Hb complexes that is mainly expressed on macrophages and monocytes[46–49]. Flow cytometry analysis revealed that CD163 was also expressed in a subset of HSCs, and its frequency decreased after PHZ treatment (Supplementary Fig. 3g). These findings might suggest that HSCs avoided radical attacks by decreasing the expression of heme and Hb receptors, which enhanced megakaryopoiesis.

### Cholesterol metabolism alters under acute anemia induction
To further elucidate the key mechanisms triggering enhanced erythroid differentiation of HSCs under acute anemia conditions, we compared the gene expression profiles of HSCs 1 day after PHZ treatment (Fig. 4a). Differentially expressed genes included cholesterol metabolism-related genes, such as VLDL receptor (*Vldlr*) (Fig. 4b). Moreover, GSEA indicated that cholesterol metabolism was altered upon PHZ injection (Fig. 4c). Therefore, we performed PB lipoprotein profiling in mice injected with PHZ. The assay revealed a large decrease in VLDL-bound triglycerol (VLDL-TG) and a significant reduction in the size of VLDL 1 day after PHZ injection, while HDL showed modest changes in particle size (Fig. 4d, e). These changes were not observed on day 3, suggesting that the altered lipoprotein profile was transient. As *Vldlr* expression was significantly higher in the HSCs of PHZ-treated mice, we investigated whether cell surface Vldlr expression was upregulated upon PHZ injection. Flow cytometry analysis revealed that Vldlr was expressed on all HSCs, and mean expression levels tended to increase upon acute anemia induction, but the difference was insignificant (Fig. 4f). To investigate whether the expression of Vldlr was correlated with differentiation potential, HSCs were sorted into two populations: Vldlr$^{high}$ and Vldlr$^{low}$ HSCs (Fig. 4f). CFU-FACS showed that Vldlr$^{high}$HSCs generated colonies containing more erythroid cells than Vldlr$^{low}$HSCs, and this difference increased upon PHZ treatment (Fig. 4g), although the number of erythroid-containing colonies (GEMMk, EMk, and GEM) did not change (Supplementary Fig. 4). These data indicated that acute anemia induction triggered a rearrangement of the lipoprotein profile, specifically the amount and size of VLDL in the blood, which correlated with the erythroid potential of HSCs possibly through Vldlr expression.

### Vldlr$^{high}$HSCs have different chromatin accessibility
Since lineage commitment and fate change of stem cells, including HSC, can be affected by epigenetic remodeling of *cis*-regulatory elements[50], we next performed Assay for Transposase-Accessible

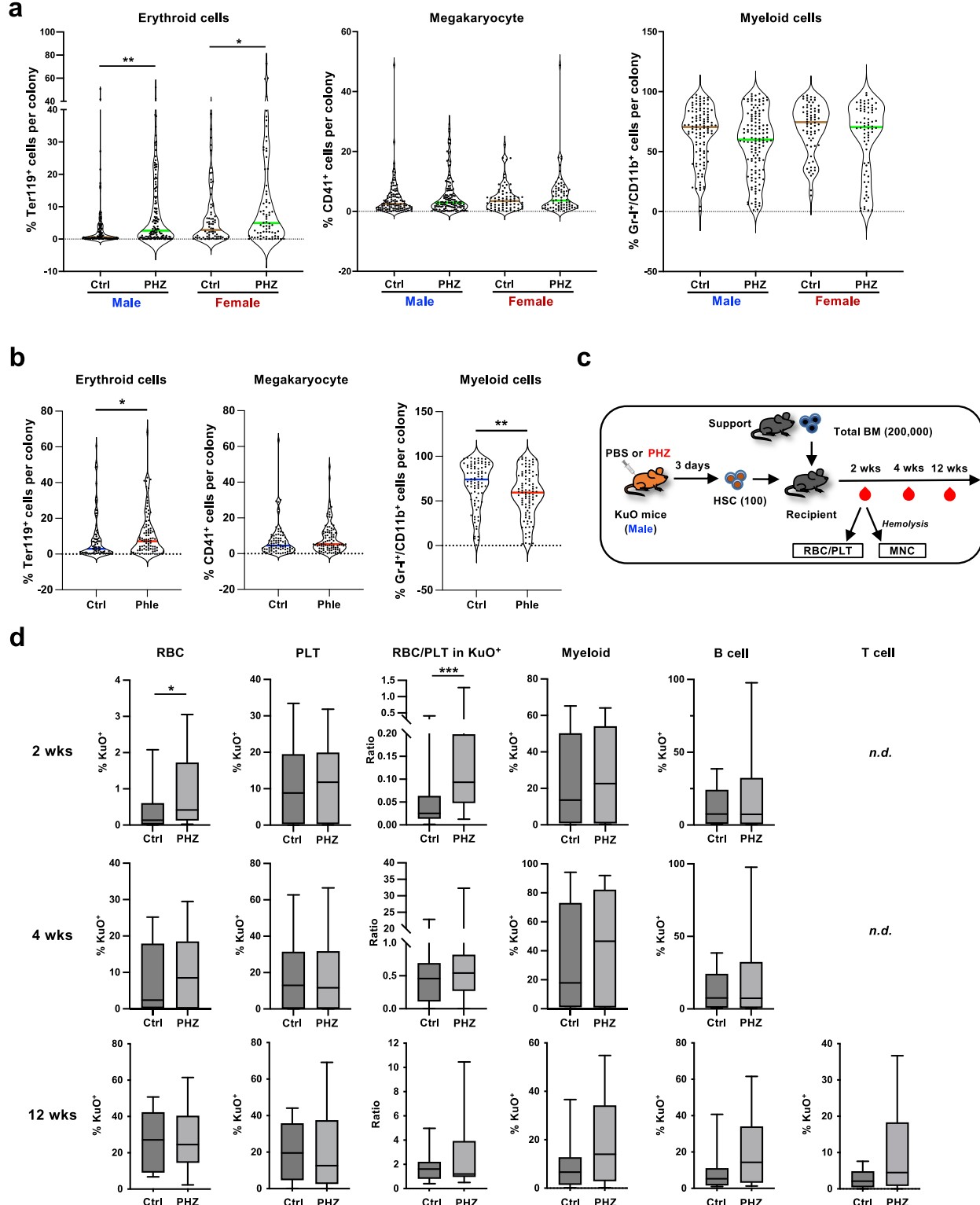

**Fig. 2 | HSCs under anemic conditions show enhanced erythroid differentiation potential. a** CFU-FACS of HSCs derived from PBS-treated control (Ctrl) or PHZ-treated mice. Data of male and female mice on day 3 are shown. Frequencies of Ter119⁺ cells (Erythrocyte), CD41⁺ cells (Megakaryocyte), and Gr-I⁺Mac-I⁺ cells (Myelocyte) in individual colonies. Each dot represents one colony. *n* = 70-130 from the total of 3 experiments. Adjusted *p* values were obtained using one-way ANOVA (Tukey's multiple comparison test). **b** CFU-FACS of HSCs derived from untreated (Ctrl) or phlebotomized male mice on day 3. *n* = 84–93 from the total of 3 experiments. *P* values were obtained using Kolmogorov–Smirnov test. **c** Experimental design of transplantation assay. One hundred CD150⁺CD34⁻KSL

cells isolated from KuO mice (donor) treated with PBS or PHZ were mixed with 2 × 10⁵ total bone marrow cells of Ly-5.2 mice (competitor) and then transplanted into Ly-5.2 mice irradiated with 900 cGy (recipient). PB was collected and subjected to flow cytometry analyses two, four, and twelve weeks after transplantation. The figure was created by authors. **d** Frequencies of KuO⁺ cells in different cell types of PB. *nd*: not detected. Twenty-one recipient mice transplanted from three donor mice for each condition were used. Box plots show the median (center line) first and third quartiles (box limits), and whiskers extend to minimum and maximum values. *P* values were obtained using two-tailed Mann–Whitney rank tests. **p* < 0.05, ***p* < 0.01, ****p* < 0.001. Exact *P* values are provided as Source Data.

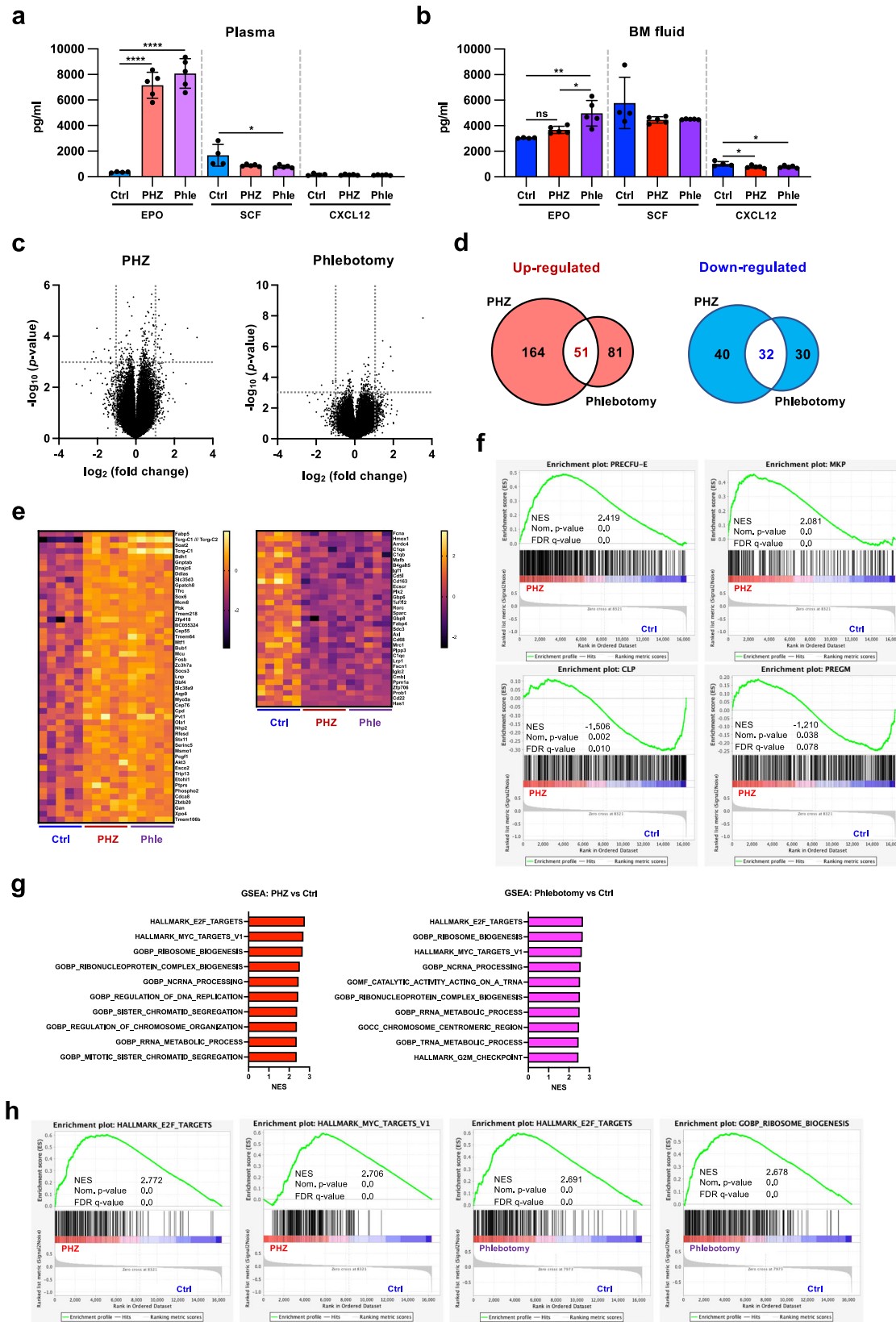

Chromatin Sequencing (ATAC-seq)[51] to compare chromatin accessibility between Vldlr[high] and Vldlr[low] HSCs under steady-state conditions and upon PHZ treatment. Principal component analysis (PCA) showed that the chromatin accessibility patterns of Vldlr[high]HSCs and Vldlr[low]HSCs were distinct, and the difference increased upon PHZ treatment (Fig. 5a). ATAC-seq identified 2,659 and 2,159 accessible

chromatin regions in steady-state Vldlr[low]HSCs and Vldlr[high]HSCs, respectively (Fig. 5b). In PHZ-treated animals, however, the difference in the number of accessible regions became more significant, as 11,326 regions were more accessible in Vldlr[low]HSCs than in Vldlr[high]HSCs, whereas 1,459 regions were more accessible in Vldlr[high]HSCs (Fig. 5c). A heatmap analysis of the accessible regions suggested that upon PHZ

**Fig. 3 | HSCs under acute anemia elevates erythroid signature genes.** ELISA assay to measure concentrations of EPO, SCF, and CXCL12 in blood plasma (**a**) or BM fluid (**b**) on day 3. Four mice for the control condition and five mice for PHZ and phlebotomy were used. Data are presented as mean values mean ± SD. Adjusted *p* values were obtained using one-way ANOVA (Tukey's multiple comparison test). **c** Volcano plot showing differentially expressed genes in HSC upon PHZ treatment (day 3) (left) and phlebotomy (right). Significant difference was defined as *p* < 0.001 and log$_2$ fold change < −1 or > 1. The *p* values are not adjusted. **d** Venn diagrams depicting the overlap of up- or down-regulated genes between PHZ-treatment and phlebotomy. **e** Heatmaps showing the genes commonly differentially upregulated (left) and downregulated (right) in HSCs (CD150$^+$CD34$^-$KSL) isolated from PHZ-injected mice (PHZ) and phlebotomized mice (Phle) compared with HSC from control mice (Ctrl). **f** Enrichment of progenitor/precursor gene signatures in HSCs of PHZ treated mice. The gene signatures of each cell type were generated from the gene expression data of Pronk et al.[76–81]. NES: normalized enrichment score. **g** Summary of GSEA comparing PHZ and Phle vs Ctrl. The top 10 gene signatures are shown. See also Supplementary Data 2. **h** Representative MsigDB enriched in the HSCs isolated from PHZ-treated or phlebotomized mice. *P* values in **c**, **f**, and **h** were adjusted with Benjamini–Hochberg method. *$p$ < 0.05, **$p$ < 0.01, ***$p$ < 0.001, ****$p$ < 0.0001. Exact *P* values are provided as Source Data.

injection, many chromatin regions were closed in the Vldlr$^{high}$HSCs (Fig. 5d). Re-mapping of chromatin regions that were not different between Vldlr$^{low}$ and Vldlr$^{high}$ HSCs under steady-state conditions showed that these were closed in Vldlr$^{high}$HSCs upon PHZ treatment, but remained accessible in Vldlr$^{low}$HSCs (Supplementary Fig. 5), indicating that the main reason for the difference in chromatin accessibility was owing to the response to acute anemia, which was exclusive for the Vldlr$^{high}$HSCs. GSEA revealed that long-term HSC-related genes were significantly enriched in newly closed chromatin regions (Fig. 5e). Additionally, megakaryocyte progenitor (MkP)-related genes were enriched in the newly closed regions (q-value 0.1), suggesting that Vldlr$^{high}$HSCs diminished their megakaryopoietic potential and simultaneously started differentiating toward erythroid cells upon PHZ treatment. Moreover, we performed motif analysis using differentially accessible regions and found that the ETS-related gene (Erg)[52] binding motif was most highly enriched in the chromatin regions that were more accessible in Vldlr$^{low}$HSCs than in Vldlr$^{high}$HSCs (Fig. 5f). Furthermore, the Erg-binding motif was also enriched in the chromatin regions closed upon PHZ treatment in both Vldlr$^{low}$HSCs and Vldlr$^{high}$HSCs (Fig. 5g). Erg is an essential regulator of HSC function and megakaryopoiesis[53–57]. It has been indicated that Erg and Fli1 form a complex with other transcription factors, including Tal1, Gata2, Runx1, Lyl1, Ldb1, and Lmo2, which binds to its transcription factor binding site in hematopoietic stem and progenitor cells (HSPCs). During erythroid commitment/differentiation, Erg/Fli1 detaches as opposed to during megakaryopoiesis[58]. Importantly, none of those transcription factors changed their transcription levels upon acute anemia induction (Figs. 3c, 3e, 4a, 4b). These results indicated that Vldlr$^{high}$HSCs and Vldlr$^{low}$HSCs were epigenetically distinct and that the number of accessible regions containing the Erg-binding motif were decreased in Vldlr$^{high}$HSCs, which was further reduced upon acute anemia induction.

## ApoE controls the erythroid potential of HSCs

Lipoproteins contain various types of apolipoproteins, and ApoE is one of those lipoproteins found mainly in VLDL, but not in LDL[59,60]. As both the concentration of VLDL-TG and the size of VLDL significantly decreased upon PHZ injection (Fig. 4d), we wondered whether ApoE expression increased upon acute anemia induction. ELISA showed that ApoE levels rapidly and transiently increased in both PB and BM fluids (Fig. 6a). To determine whether ApoE induction is necessary for erythroid-biased differentiation of HSCs upon anemia induction, ApoE-knockout (*Apoe$^{-/-}$*) mice[61] were injected with PHZ, and the differentiation potential of HSCs was assessed using CFU-FACS. We found that the erythroid output did not increase in *Apoe$^{-/-}$*HSCs (Fig. 6b). Of note, *Apoe* is highly expressed in HSCs according to Gene Expression Commons (https://gexc.riken.jp)[62] (Supplementary Fig. 6a). To exclude the possible impact of endogenous *Apoe* depletion on the alleviated HSC response, we transplanted total BM cells of WT mice to lethally irradiated WT or *Apoe$^{-/-}$* recipient mice. Three months later, acute anemia was induced in the engrafted mice, and CFU-FACS was performed by sorting donor-derived HSCs (Fig. 6c). The assay showed a similar result: no increase in the erythroid differentiation potential in HSCs engrafted to *Apoe$^{-/-}$* mice upon PHZ treatment, while HSCs engrafted to WT mice showed a significantly higher erythroid content (Fig. 6d). This result confirmed that lack of the response against acute anemia induction was due to the loss of environmental ApoE but not the endogenous *Apoe* expression in HSCs.

To analyze whether the impaired ApoE signal affects erythroid generation in vivo, we injected WT or *Apoe$^{-/-}$* mice with PHZ and analyzed their PB 7 days post PHZ treatment. As a result, blood profiling did not find significant differences; however, we saw trends that RBC profiles (RBC number, HCT, HGB) were lower in PB of *Apoe$^{-/-}$* than of WT mice (Fig. 6e). We also analyzed frequencies of different blood lineages in the BM and spleen of the PHZ-injected WT and *Apoe$^{-/-}$* mice using flow cytometer. We observed that the frequencies of erythroid cells were lower in *Apoe$^{-/-}$* mice while other cell types (myeloid, B cells, and T cells) were rather higher (Fig. 6f, g). These results suggest active contribution of ApoE in stress erythropoiesis in vivo.

Next, we investigated whether ApoE functions as a trigger for the enhanced erythroid differentiation of HSCs. Quantitative reverse transcription-PCR (qRT-PCR) analysis revealed that wild-type (WT) HSCs cultured with recombinant ApoE for three days expressed higher levels of *Fabp5* (Fig. 6h), which was also upregulated in the HSCs of PHZ-treated mice (Fig. 3e). CFU-FACS of HSCs pre-cultured for three days with ApoE generated more erythroid cells (Fig. 6i). To ask whether Vldlr in HSCs is needed for the acute anemia response, we tested knockdown the expression of *Vldlr* using short-hairpin RNA (shRNA). As a result, we observed that Vldlr-knockdown alleviated the increase in erythroid potential upon PHZ-injection in the infected HSCs while shLuciferase control cells still showed higher erythroid content (Supplementary Fig. 6b).

## ApoE modifies chromatin accessibility in HSCs

To unravel the genetic changes in HSCs following ApoE treatment, we performed ATAC-seq. ApoE-treated HSCs showed a distinct chromatin accessibility pattern compared with uncultured HSCs or control HSCs (Fig. 7a, b). Motif analysis revealed that regions containing the Erg-binding motif were one of the most closed regions upon ApoE treatment (Fig. 7c). We compared the gene expression profiles of Erg$^{+/-}$ HSCs[57] and HSCs of PHZ-injected mice (day 3) (Fig. 3). GSEA showed that genes upregulated upon PHZ treatment were enriched in Erg$^{+/-}$ HSCs, whereas genes downregulated upon PHZ treatment were enriched in WT HSCs (Fig. 7d). To test whether the transient suppression of Erg function enhances erythropoiesis in HSC, we treated HSCs with 1-(2-Thiazolylazo) −2-naphthol, an Erg-inhibitor (ERGi), and analyzed their erythroid potential using CFU-FACS. The result showed that ERGi treatment significantly increased the frequency of erythroid cells in each colony, similar to recombinant ApoE (Fig. 7e). These findings indicated that ApoE could be critical regulator that enhances the erythroid differentiation potential of HSCs, potentially by closing the Erg-binding sites.

## Discussion

EPO stimulates the proliferation and differentiation of erythroid progenitor/precursor cells; it is therefore believed that stress responses in acute anemia are mainly governed by the committed progenitor and/or precursor levels, but the contribution of HSCs is unclear. In the present study, we demonstrated that HSCs respond to acute anemic stress

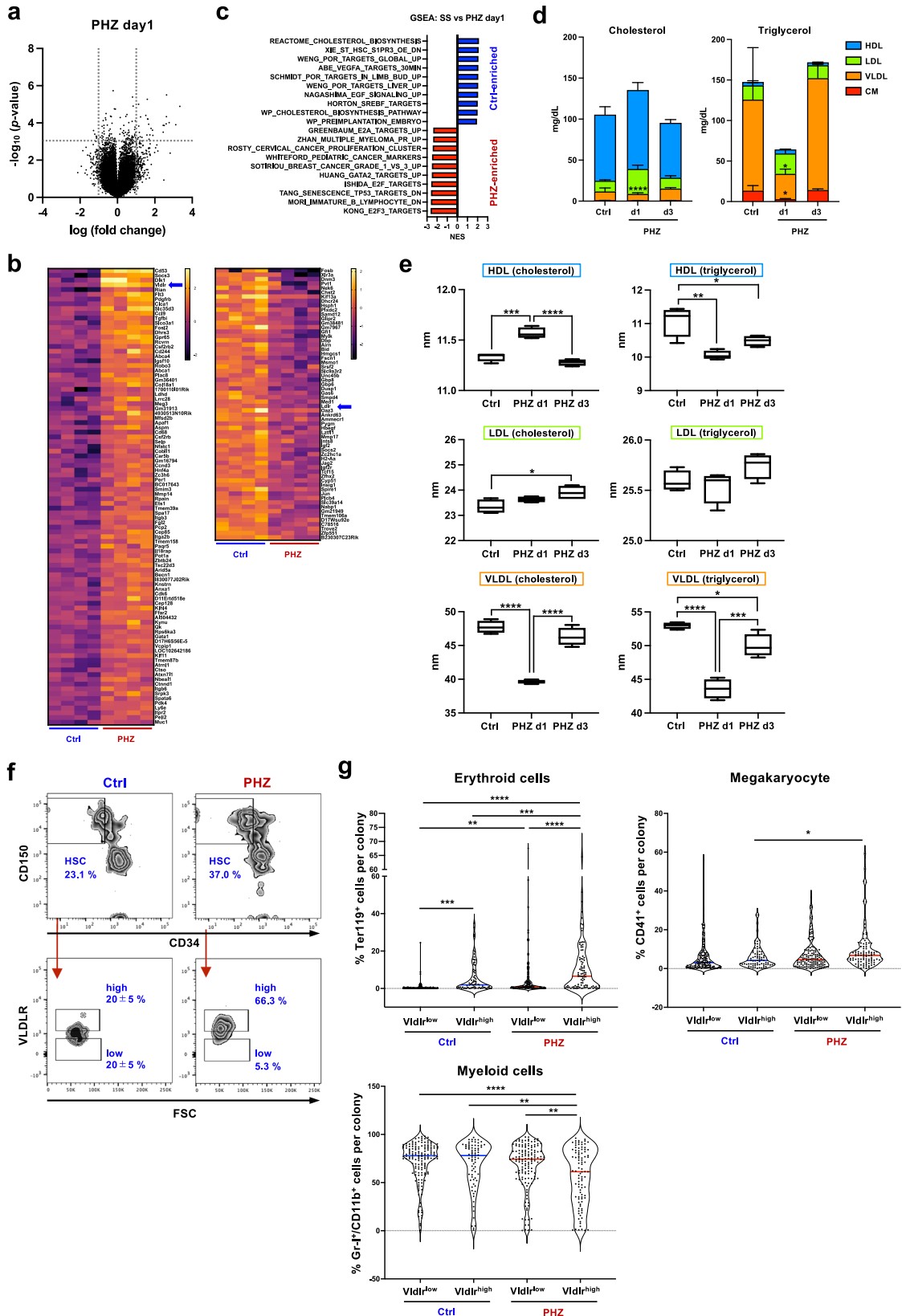

independently of the effects of EPO. To evaluate the lineage potential of individual HSC in vitro, we developed a quantitative CFU assay, CFU-FACS (Supplementary Fig. 2). This assay enabled us to quantitatively compare the lineage potential of HSCs in vitro, as opposed to categorizing them into a limited number of colony types. We demonstrated that HSCs under acute anemia conditions showed a great potential to

differentiate into erythroid cells. Previous studies have mainly focused on the effects of EPO on hematopoietic stem and progenitor cells[17,19]. In addition to publications reporting the expression of EPO-R in HSPCs, Pradeep et al. demonstrated that EphB4 expressed on the cell surface can function as an EPO receptor in tumor cells[63], indicating that the expression pattern of EPO-R may not define the target populations of

**Fig. 4 | Cholesterol metabolism is altered upon acute anemia induction.**
**a** Volcano plot showing differentially expressed genes in HSCs 24 hours after PHZ treatment. Significant difference was defined as $p < 0.001$ and $\log_2$ fold change $< -1$ or $> 1$. The $p$ values are not adjusted. **b** Heatmaps showing up- or down-regulated genes in HSCs isolated from PHZ-injected mice (PHZ) compared with HSCs from control mice (Ctrl). The blue arrows highlight *Vldlr* and *Ldlr* genes. **c** Summary of GSEA comparing PHZ day1 vs Ctrl HSC. The top 10 gene signatures are shown. NES: normalized enrichment score. See also Supplementary Data 3. **d** Lipoprotein profiling of PB. The concentrations of cholesterol and triglycerol in different lipoproteins are shown. HDL high-density lipoprotein; LDL, low-density lipoprotein; VLDL, very low-density lipoprotein; CM, chylomicron. Four mice for each condition were used. Data are presented as mean values + SD. **e** The particle size of each lipoprotein. Four mice for each condition were used. Box-plots show the median (center line) first and third quartiles (box limits), and whiskers extend to minimum and maximum values. **f** Vldlr expression on HSCs. Representative FACS plot of Vldlr expression on CD150$^+$CD34$^-$ population (KSL gated) of Ctrl and PHZ-treated mice are shown. Vldlr gates show Vldlr$^{low}$ and Vldlr$^{high}$ populations. **g** CFU-FACS of Vldlr$^{high}$HSC and Vldlr$^{low}$HSC derived from Ctrl and PHZ-treated mice. $n = 75$-$151$ from the total of 3 experiments. Adjusted $p$ values were obtained using one-way ANOVA (Tukey's multiple comparison test). $*p < 0.05$, $**p < 0.01$, $***p < 0.001$, $****p < 0.0001$. Exact $P$ values are provided as Source Data.

EPO. In this study, we analyzed HSC-derived colonies formed in semi-solid medium containing a sufficient concentration of EPO, indicating that the impact of EPO on erythroid differentiation is equal under different cellular conditions. However, we observed a significantly higher erythroid potential in HSCs derived from anemic mice (Fig. 2a, b). This suggests that acute anemia induction triggers alterations in cellular characteristics that cannot be overcome through EPO signaling. Yet, our microarray data found significant upregulation of *Tfrc* and *Socs3*, well-known downstream target genes of the EPO signal (Fig. 4b). These genes are not induced by the addition of ApoE to the in vitro HSC culture (data not shown) and we have not identified exact cytokine(s) and their signals inducing these genes; it would be an important future study to discover additional mechanism stimulating those cytokine-related signaling cascades. For instance, *Tfrc* is known to be induced by iron-deficiency and hypoxia[64,65].

We also found that male and female mice responded differently against acute anemic stress; the HSC expansion in the BM was more pronounced in male mice while HSC expansion happened in spleen of female mice (Fig. 1e, f). How the sex difference is made is unclear, however, it has been suggested that bone morphogenic protein (BMP) signaling which has been implicated in the splenic stress erythropoiesis is affected by sex difference[66–68]. These observations indicate a potential mechanism explaining why male and female spleen differentially responded to stress erythropoiesis, possibly through BMP signals.

Gene expression analyses revealed the transient involvement of cholesterol and lipoprotein metabolism after the induction of acute anemia. Lipoproteins are central to lipid and cholesterol metabolism. LDL and HDL have been implicated in HSPC regulation. In particular, HDL is known to suppress the proliferation of HSCs via ATP-binding cassette transporters[29]. In addition, hypercholesterolemia induces the mobilization of BM cells[69–71]. Thus, different lipoproteins, including apolipoproteins, have distinct roles and functions in hematopoiesis. Lipoproteins contain various types of apolipoproteins, and ApoE is one of the lipoproteins found mainly in VLDL, but not in LDL[59,60]. Although we did not measure the lipoprotein profiles in the BM, PB profiles clearly showed a decrease in the concentration of VLDL-TG and the particle size of VLDL (Fig. 4d, e). We demonstrated that VLDL and ApoE play key roles in the response to acute anemia; the addition of recombinant ApoE increased the erythroid output of HSCs (Fig. 6d), whereas HSCs of *Apoe*$^{-/-}$ mice did not exhibit enhanced erythroid potential (Fig. 6b). Importantly, *Apoe*$^{-/-}$ mice have normal HSC frequency and cell cycle status under steady-state conditions when fed a normal diet[28,72] while erythropoiesis is significantly enhanced when fed with high-fat diet[73], suggesting that ApoE plays a unique role in HSCs under stress erythropoiesis conditions which is affected by lipid metabolism. In fact, we saw an indication that recovery of RBC profiles was delayed in ApoE KO (fed with normal diet) (Fig. 6e–g). It is challenging to accurately evaluate how much of the impaired recovery were affected by the altered HSC regulation due to ApoE depletion, as RBCs are generated not directly from HSCs but rather from their immediate progenitors and precursors; thus, impaired erythroid commitment/differentiation in ApoE KO HSCs was presumably compensated or masked therefore the differences were modest. Furthermore, we found a correlation between cell surface Vldlr expression and the erythroid potential of HSCs. Vldlr$^{high}$HSCs and Vldlr$^{low}$HSCs were functionally distinct even under steady-state conditions to a modest degree; however, the difference became larger upon acute anemia induction, as Vldlr$^{high}$HSCs after acute anemia generated a significantly higher proportion of erythroid cells (Fig. 4g).

How fate commitment and lineage choice happen is a fundamental question. Importantly, ATAC-seq revealed that chromatin accessibility did not increase in Vldlr$^{high}$HSCs after acute anemia induction, instead, the chromatin regions that mainly regulate HSC programming and megakaryopoiesis were closed. Moreover, we found that the Erg-binding motif was enriched in these closed chromatin regions. Erg is a transcription factor belonging to the ETS family[52], and is considered an essential regulator of HSC function and activity, as its genetic depletion in mice leads to exhaustion of functional HSCs[53,56,57]. In addition, Erg, along with other transcription factors such as Fli1 and Ets2, promote HSC function and megakaryopoiesis[54,55]. Several studies have suggested links between HSC capacity and megakaryopoiesis potential[36,74]. HSC- and modestly megakaryocyte-related gene signatures were reduced upon acute anemia induction (Fig. 5e). These results indicate that, unlike the typical differentiation control in which transcription factors induce lineage-specific gene expression, enhanced erythropoiesis can result from the reduced megakaryopoiesis potential present in HSCs. However, it is undiscovered how the chromatin accessibility particularly target regions of Erg is reduced upon acute anemia induction without alterations in the Erg expression level. Recent studies have proposed that small lipid molecules such as short chain fatty acids and cholesterols can bind to chromatin and regulate gene expression[75]. Since we have seen the induction of Fabp5, a protein transporting fatty acid (Figs. 3e and 6c), it is interesting to study whether fatty acids bind to chromatin and affect chromatin accessibility (or transcription activity of Erg protein directly).

Importantly, epigenetic changes upon acute anemia induction were selectively observed in Vldlr$^{high}$HSCs; this subfraction of HSCs starts differentiating upon ApoE stimulation, while the counter fraction (Vldlr$^{low}$HSCs) remained undifferentiated to maintain the HSC pool. Hence, this might be a mechanism that enables both a rapid supply of desired cell types and the maintenance of undifferentiated HSCs. We previously reported that a subfraction of HSCs expressing cell surface molecule Jam2 (Jam2$^{high}$HSCs) preferentially differentiates toward T cell lineages, and the differentiation potential is enhanced by increased Notch signaling upon binding of Jam2 to its target protein Jam1[76]. Both Vldlr and Jam2 are expressed in all HSCs but at different levels. Presumably, such gradual differences in protein expression create a wide variety of cellular responses with HSC heterogeneity.

In conclusion, we discovered a mechanism of erythroid differentiation in HSCs, involving lipoprotein changes leading to epigenetic changes. Our findings suggest that lineage commitment can be governed without the implications of instructive cytokine signals or signal transduction.

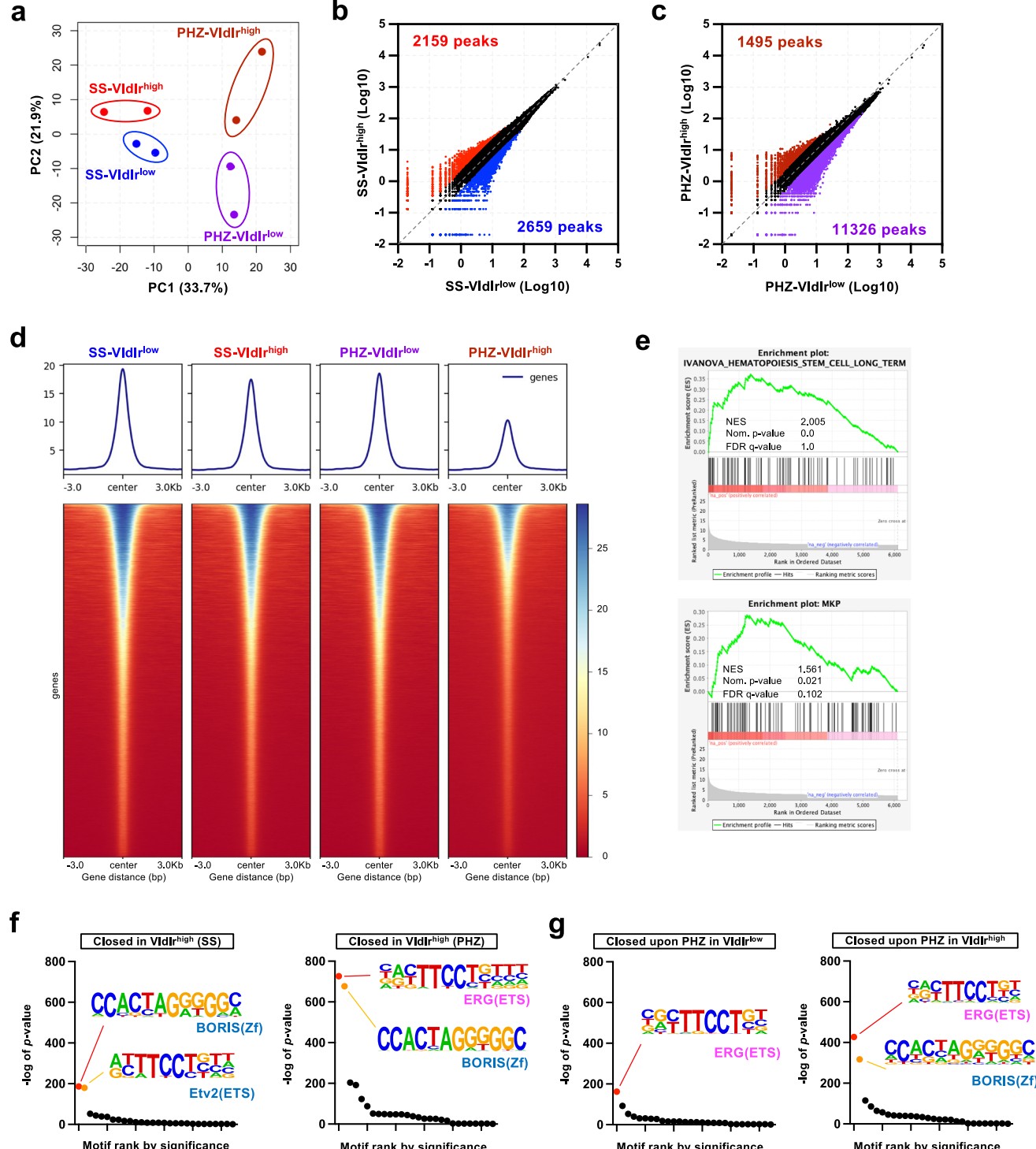

**Fig. 5 | Chromatin accessibility difference between Vldlr^high HSCs and Vldlr^low HSCs. a** Principal component analysis (PCA) of the ATAC-seq data of Vldlr^high and Vldlr^low HSC derived from the steady state (SS) or PHZ-treated (PHZ) mice. Data are presented as mean values +/− SD. Scatter plots comparing ATAC-seq peaks between Vldlr^high HSC and Vldlr^low HSC in the SS mice (**b**) or PHZ-treated mice (**c**). **d** Average plot (top) and heatmap (bottom) of open chromatin regions in Vldlr^high and Vldlr^low HSC derived from the steady state (SS) or PHZ-treated (PHZ) mice. **e** The enrichment of genes related to "unchanged between Vldlr^high and

Vldlr^low HSC under SS conditions but closed in Vldlr^high HSC upon PHZ treatment". *P* values were adjusted with Benjamini–Hochberg method. **f** Motif analysis of open chromatin regions closed in Vldlr^high HSC under SS condition (left) or after PHZ-treatment (right). Ranks of motifs enriched in each condition are shown. *P* values were not adjusted. **g** Motif analysis of open chromatin regions closed upon PHZ treatment in Vldlr^low HSC (left) or Vldlr^high HSC (right). Ranks of motifs enriched in each condition are shown. *P* values were not adjusted. *$p < 0.05$, **$p < 0.01$, ***$p < 0.001$, ****$p < 0.0001$. Exact *P* values are provided as Source Data.

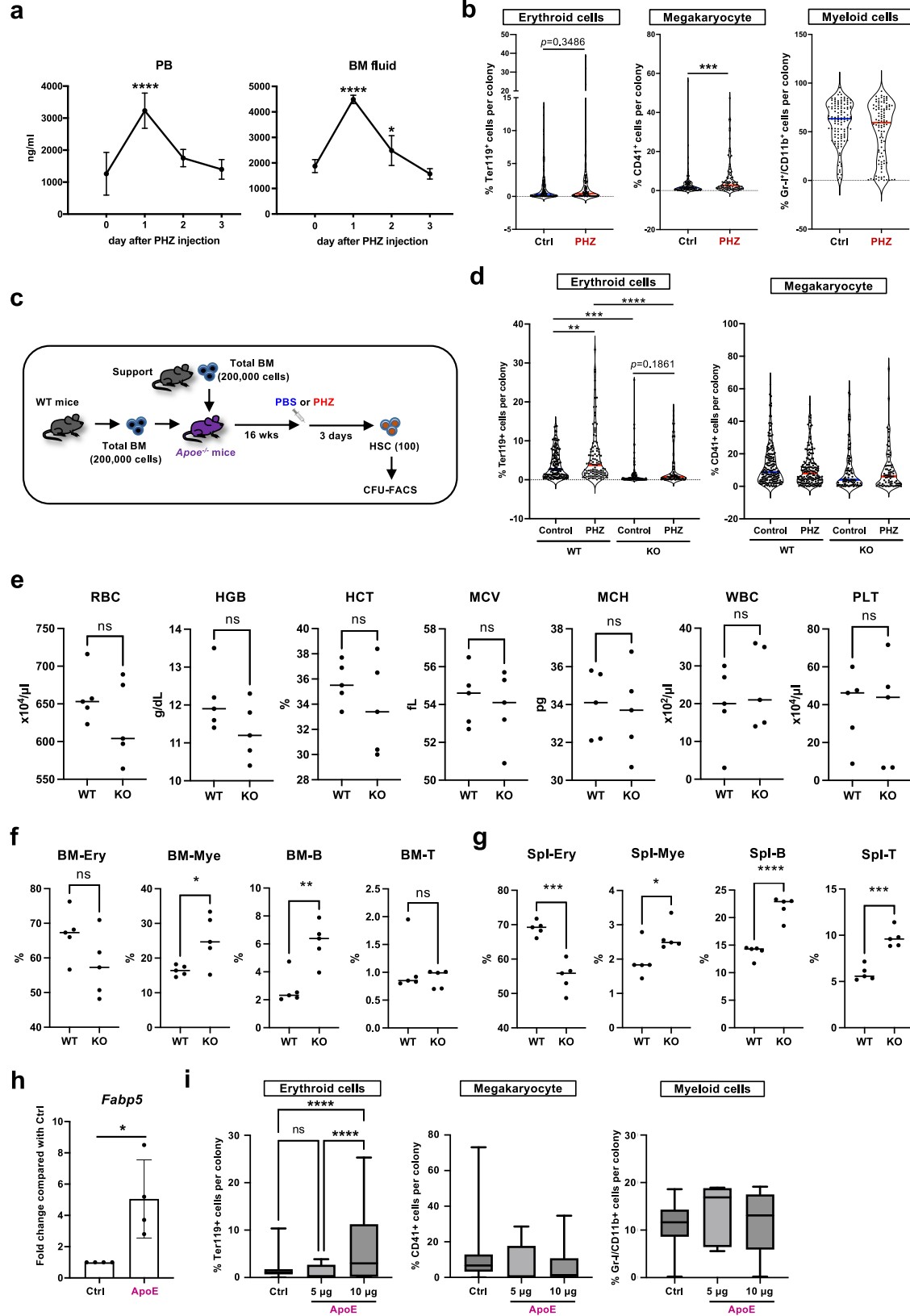

## Methods

### Mice

All experiments were approved by the Lund University Animal Ethical Committee, Swedish Board of Agriculture, and Animal Research Facility of Kumamoto University guidelines. Young C57BL/6 J (Ly-5.2) male mice were obtained from Janvier Labs. All animals were maintained in individually ventilated racks and given autoclaved food and water ad libitum. Kusabira Orange (KuO) mice[35] were gifted from Hiromitsu Nakauchi's laboratory at Tokyo University, Japan. ApoE knockout mice (B6.129P2-*Apoe*[tm1Unc]/J) and Ki-67 reporter mice (*Mki67*[tm1.1Cle/J])[72] were purchased from The Jackson Laboratory.

**Fig. 6 | ApoE controls the erythroid differentiation potential of HSCs. a** ELISA assay to measure concentrations of ApoE in blood plasma (left) or BM fluid (right). Five mice were used for each time point. The fold changes to day 0 were evaluated. Adjusted *P* values were obtained using one-way ANOVA (Dunnett's multiple comparisons test). **b** CFU-FACS of HSCs derived from Ctrl or PHZ-treated *Apoe*$^{-/-}$ mice. *n* = 137–159 from the total of 3 experiments. *P* values were obtained using two-tailed Mann–Whitney rank tests. **c** Experimental design of CFU-FACS of HSCs derived from *Apoe*$^{-/-}$ mice engrafted with WT BM cells. WT BM cells were transplanted to WT mice or *Apoe*$^{-/-}$ mice, and three months later the mice were injected with PHZ or PBS. CFU-FACS were performed using re-isolated donor-derived HSCs. Data from the total of 3 experiments. The figure was created by authors. **d** Result of CFU-FACS of HSCs. *n* = 77–168 colonies from three independent recipient mice. Adjusted *p* values were obtained using one-way ANOVA (Tukey's multiple comparison test). **e** PB analysis of the PHZ-treated WT mice or *Apoe*$^{-/-}$ mice. RBC, HGB, hematocrit (HCT), mean corpuscular volume (MCV), mean corpuscular hemoglobin (MCH), white blood cell count (WBC) and platelet count (PLT) on day 7 are shown. Five

mice for each genotype were used. *P* values were obtained using two-tailed unpaired *t* tests. **f, g** BM and spleen (SPL) analysis of the PHZ-treated WT mice or *Apoe*$^{-/-}$ mice. Ter119$^+$ erythroid cells (Ery), CD11b$^+$/Gr-I$^+$ myeloid cells (Mye), B220$^+$ B cells (B), and CD4$^+$/CD8$^+$ T cells (T) on day 7 are shown. Five mice for each genotype were used. *P* values were obtained using two-tailed unpaired *t* tests. **h** qRT-PCR for *Fabp5* expression in HSC treated with recombinant ApoE for 3 days. Relative expression compared to the DMSO control (Ctrl) is shown. The expression levels were normalized to *Hprt*. Four individual experiments were done. *P* value was obtained using two-tailed unpaired *t* test. **i** CFU-FACS of HSC treated with recombinant ApoE for 3 days. One-hundred-seven (Ctrl), ninety-five (5 μg), or eighty-one (10 μg) colonies from the total of three experiments were analyzed. Box plots show the median (center line) first and third quartiles (box limits), and whiskers extend to minimum and maximum values. Adjusted *p* values were obtained using one-way ANOVA (Tukey's multiple comparison test). *\*p* < 0.05, \*\**p* < 0.01, \*\*\**p* < 0.001, \*\*\*\**p* < 0.0001. Exact *P* values are provided as Source Data.

### Acute anemia induction
Hemolytic anemia was induced by intraperitoneally injecting 60 mg/kg of phenylhydrazine (PHZ, Sigma-Aldrich). Alternatively, acute anemia was induced by phlebotomy; approximately 300 μl of peripheral blood was collected from the tail vein of the mouse.

### Peripheral blood analysis
Peripheral blood samples were collected from the tail vein to EDTA-coated microcuvette tubes (Sarstedt), and blood profiles were analyzed on a KX-21N hematology analyzer (Sysmex).

### Flow cytometry
Adult BM cells were isolated by crushing tibias, femurs, and iliac bones of 8 to 10 weeks old mice with a mortar and pestle in PBS. Mononuclear cells were collected from the buffy coat after Lymphoprep™ separation. C-Kit positive (c-Kit$^+$) cells were enriched using a magnetic separation system (MACS) with anti-c-Kit magnetic beads (Miltenyi Biotec). The enriched cells were stained with the following antibodies conjugated to FITC, PE, PE-Cy5, PE-Cy7, APC, APC-eFluor780 or Brilliant Violet: anti-Vldlr (Abcam), -CD163 (TNKUPJ, eBiosciences), -CD34 (RAM34, BD), -CD150 (TC15-12F12.2, BioLegend), -CD16/32 (93, BioLegend), -c-Kit (2B8, eBioscience), -Sca-1 (D7, BioLegend), -CD127 (A7R34, BioLegend), -CD45.1 (A20, BioLegend), -CD45.2 (104, BioLegend), -CD3 (145-2C11, BioLegend), -CD4 (H129.19, BD), -CD8 (53-6.7, BioLegend), -Gr-1 (RB6-8C5, BioLegend), -CD11b (M1/70, BioLegend), -B220 (RA3-6B2, BioLegend), and Ter119 (TER119, BioLegend). Anti-CD3, -B220, -CD11b, -Gr-I, and Ter119 antibodies were used as lineage antibody mix. Dead cells were excluded using 7-Amino-Actinomycin-D (7AAD) staining. Cells were sorted on FACS Aria III or analyzed on FACS LSRII or LSR Fortessa (BD). Collected data were analyzed on the FlowJo software (Tree Star).

### Cell cycle assay
Total BM cells isolated from Ki-67 reporter mice treated with or without PHZ were stained with HSC markers, and Ki67 expression levels were analyzed on FACS AriaIII (BD). Ki-67 RFP$^+$ cells were defined as cycling cells.

### In vitro culture
One hundred CD34$^-$CD150$^+$KSL cells were directly sorted into 96 well plates containing StemSpan® SFEM media containing 10 ng/ml mSCF and 100 ng/ml hTPO. The sorted cells were cultured with the following soluble factors: Haptoglobin (Lee Biosolutions), Hemopexin (Sino Biological), ApoE (Abcam), 1-(2-Thiazolylazo)−2-naphthol (ERGi, Sigma-Aldrich).

### CFU-FACS
One hundred CD150$^+$CD34$^-$KSL cells were sorted into StemSpan® SFEM media. The sorted cells were mixed into 3 ml of MethoCult™ GF M3434

(STEMCELL Technologies), and one-third of the mixture was plated into 35 mm dishes (x 2). The cells were cultured in a $CO_2$ incubator at 37 °C for 11 days. After the culture, formed colonies were separately picked and put into 500 μl of PBS in a microtube. The tubes were thoroughly vortexed and centrifuged at $400 \times g$ for 2 min. After removing the supernatant, TruStain FcX PLUS (S17011E, BioLegend) was added and incubated for 10 min to block non-specific binding of immunoglobulin to the Fc receptors. The cells were then stained with antibodies targeting Gr-I, CD11b, CD41, and Ter119 for 15 min. Before analysis, the pellets were diluted with 200 μl of PBS supplemented with 1% of FBS. Analysis was done using BD LSRII or FACSymphony™.

### Long-term competitive repopulation assay
A competitive repopulation assay was performed using the KuO mice. CD150$^+$CD34$^-$KSL cells isolated from KuO mice (donor) were mixed with $2 \times 10^5$ total bone marrow cells of Ly-5.2 mice (competitor) and then transplanted into Ly-5.2 mice irradiated with 900 cGy (recipient). Two, four, and twelve weeks after transplantation, PB from the tail vein of recipient mice were collected. Ten microliters (10 μl) of the collected PB were diluted into 100 μl PBS and stained with anti-CD41 and Ter119 antibodies. The rest of the PB was treated with $NH_4Cl$ for red blood cell lysis and then stained with anti-CD45.1, -CD45.2, -CD3, -Gr-1, -CD11b, and -B220 antibodies. Donor contribution (chimerism) was determined as a formula of % donor / (% donor + % competitor) x 100.

### Enzyme-linked immunosorbent assay (ELISA)
Blood plasma was collected by centrifuging peripheral blood at $1500 \times g$ for 3 min. To collect BM fluid, the femur, tibia, and iliac were isolated, and muscle and other tissue were removed thoroughly. The bones were placed in a 200 μl vial with a hole in the bottom. The 200 μl vial was placed in a 500 μl vial with a hole in the bottom which in turn was placed in a 1.5 ml vial. Fifty (50) μl of PBS was then added to the bones, and the vials were centrifuged for 5 minutes at $300 \times g$ at 4 °C. The bones were removed from the vial, and the supernatant was collected and transferred to a new vial which was centrifuged for 20 min at $2000 \times g$ at 4 °C. The cell- and debris-free supernatant was collected and stored at −80 °C. The concentrations of each cytokine were measured using the following kits: Haptoglobin, DuoSet® Mouse Haptoglobin ELISA Kit (R&D); SCF, Mouse SCF ELISA kit (Abcam); CXCL12, Mouse SDF-1 alpha CXCL12 ELISA (Thermo); EPO, Quantikine® Mouse Epo (R&D); ApoE, Mouse Apolipoprotein E ELISA kit (Abcam); Para-meter™ Estradiol (R&D). Samples were analyzed on SPECTROstar Nano (BMG Labtech).

### Cytokine Array
Cytokine Array was performed using Mouse Cytokine Antibody Array (144 Targets) (abcam) according to the manufacturer's protocol. Briefly, blood plasma collected from untreated or PHZ-injected mice

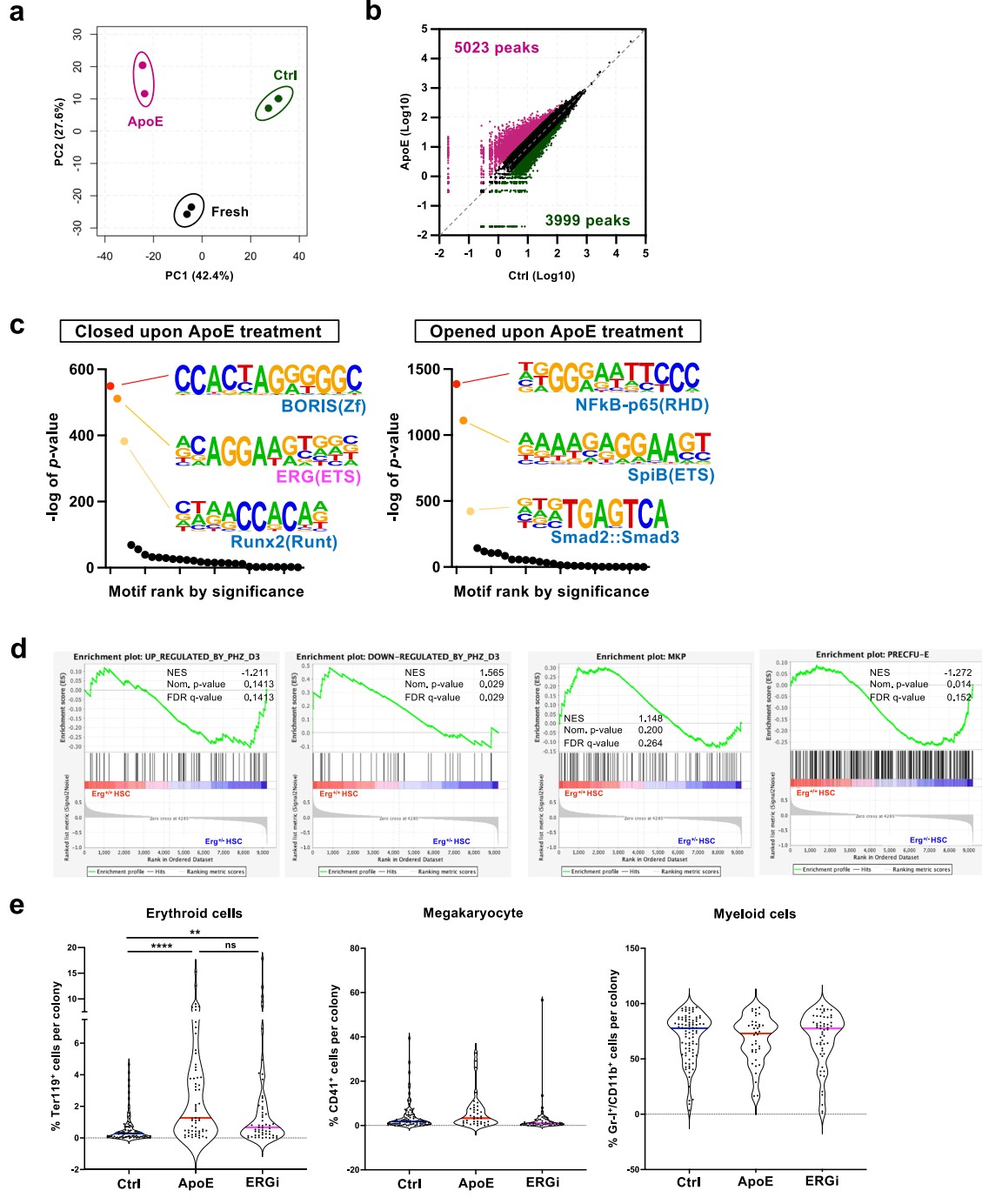

**Fig. 7 | ApoE modifies chromatin accessibility in HSCs. a** Principal component analysis (PCA) of the ATAC-seq data of freshly isolated HSC (Fresh), control culture (Ctrl), and recombinant ApoE-treated HSC (ApoE). **b** Scatter plots comparing ATAC-seq peaks between control culture (Ctrl) and recombinant ApoE-treated (ApoE) HSC. **c** Motif analysis of open chromatin regions closed (left) or opened (right) upon ApoE treatment. Ranks of motifs enriched in each condition are shown. *P* values were not adjusted. **d** The enrichment of genes related to "closed upon ApoE treatment" in *Erg*[+/-] HSC gene expression data (Knudsen et al.[57]). *P* values were adjusted with Benjamini–Hochberg method. **e** CFU-FACS of HSCs treated with recombinant ApoE or Erg inhibitor (ERGi) for 3 days. *n* = 55-89 from the total of 2 experiments. Adjusted *p* values were obtained using one-way ANOVA (Tukey's multiple comparison test). \**p* < 0.05, \*\**p* < 0.01, \*\*\**p* < 0.001, \*\*\*\**p* < 0.0001. Exact *P* values are provided as Source Data.

were incubated with the Cyrokine Array membranes for 2 hours, and the binding of each cytokine was detected by chemiluminescence.

## Microarray analysis

CD150[+]CD34[-]KSL cells were sorted from BM of 10 weeks old C57BL/6 mice treated with PBS or PHZ. Total RNA was isolated from the sorted cells using Rneasy® Micro Kit (Qiagen) according to the manufacturer's protocol. After the quality/quantity determination of the extracted RNA, cDNA was synthesized and amplified using Ovation® Pico WTA System V2 (NuGEN), or extracted RNA was amplified and converted to cDNA using GeneChip® 3' IVT Pico Reagent Kit (Affymetrix).

Fragmented and labeled double-strand cDNA were hybridized to Affymetrix Mouse Genome 430 PM Array Plates using an Affymetrix GeneTitan® system controlled by the Affymetrix GeneChip® Command Console® software v4.2 or v4.3.3. The fluorescent signals were measured with an Affymetrix GeneTitan® system controlled by the

Affymetrix GeneChip® Command Console® software v4.2 or v4.3.3. Gene level summarized probe set signals in $\log_2$ scale were calculated from Affymetrix CEL files using the RMA algorithm as implemented in the Affymetrix GeneChip® Expression Console® v1.4 Software. Sample processing was performed at a Genomics Core Facility, "KFB – Center of Excellence for Fluorescent Bioanalytics" (Regensburg, Germany; www.kfb-regensburg.de). Differential genes were called using the Limma R package using a *p*-value cut-off of 0.01 or 0.05.

## Lipoprotein profiling
The cholesterol and triglyceride profiles in plasma lipoproteins were analyzed using a gel-permeation HPLC method (LipoSEARCH®; Immuno-Biological Laboratories Co., Ltd., Akita, Japan)[73,74]. The particle sizes of each lipoprotein were determined using the retention times of the peaks observed on a chromatogram with a linear calibration curve[75].

## Quantitative RT-PCR
CD150⁺CD34⁻KSL cells were directly sorted into lysis buffer, and total RNA was isolated using Rneasy® Micro Kit. CT values were averaged, and relative expression compared to HPRT was calculated using $2^{-\Delta CT}$ formula. The following probes were used: *Fabp5*, Mm00783731_s1; *Hprt*, Mm03024075_m1. RT-PCR reactions were performed on Light-Cycler® 96 System (Roche Diagnostics).

## ATAC sequence
ATAC-seq was performed as previously described[51]. Briefly, 2500 to 5000 cells were sorted into microtubes containing PBS supplemented with 2.5 % FBS. After sorting, the cells were centrifuged at $300 \times g$ for 10 min at 4 °C to pellet down, and the supernatant was removed. The cell pellets were washed with 1 x RSB and centrifuged at $500 \times g$ for 5 min at 4 °C. After removing the supernatant, the pellets were resuspended in cold lysis buffer (10 mM Tris-HCl, 10 mM NaCl, 3 mM MgCl2, 0.01 % digitonin, 0.1 % Tween-20, and 0.1 % NP-40) and incubated for 3 min on ice. Then 1 ml of wash buffer was added and centrifuged at $500 \times g$ for 10 min at 4 °C to pellet the nuclei. The pellets were resuspended in a transposition reaction mix (2 µl transposase, 25 µl TD buffer, PBS, digitonin, Tween-20), and the transposition reaction was performed at 37 °C for 1 hour. The samples were then purified using MinElute Reaction Clean Up kit (QIAGEN). Trimmomatic, Bowtie2, MACS2, HOMER, and deepTools were used to analyze the sequence data.

## Statistical analysis
Statistical analyses were performed using Prism software (GraphPad). The one-way ANOVA residuals were tested against normality distribution in the case of more than two groups under test. The statistical details are documented in the corresponding figure legend. Statistical significance was accepted when $p < 0.05$.

## Reporting summary
Further information on research design is available in the Nature Portfolio Reporting Summary linked to this article.

## Data availability
The microarray data generated in this study have been deposited in the GEO database under accession codes GSE162408 and GSE212392. The ATAC-seq data generated in this study have been deposited in the GEO database under accession code GSE274144. Source data are provided with this paper.

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

## Acknowledgements

We thank Jun Chen and Anna Rydström for experimental support, and Jill Storry and Leal Obroglu for scientific discussions and critical reading. This work was supported by JSPS KAKENHI Grant Number JP22K16324 (K.S.), Royal Physiographic Society of Lund (K.S., M.v.d.G.), Olle Engkvist Foundation (K.S., J.H., M.L.O., J.F., K.M.), Knut and Alice Wallenberg Foundation (M.v.d.G., K.M.), O.E. Edla Foundation (M.v.d.G.), the Swedish Cancer Society (V.R., V.S., K.M.), and the Swedish Research Council (K.M.). K.M. was funded by StemTherapy program at Lund University. The Lund Stem Cell Center was supported by a Center of Excellence grant in life sciences from the Swedish Foundation for Strategic Research.

## Author contributions

K.M. designed the project. K.S., M.v.d.G., T.U., G.G., J.H., M.L.O., J.F., and K.M. planned experiments. K.S., M.v.d.G., T.U., N.M., J.S., H.S., S.K., T.M., V.S., V.R., M.S., S.N., and K.M. performed experiments. M.v.d.G., T.U., S.L., and K.M. analyzed gene expression data and ATAC-seq data. K.S., M.v.d.G., and K.M. wrote the manuscript with input from all authors.

## Funding

## Competing interests

The authors declare no competing interests.
