## [Peer Review File · Nature Communications]

REVIEWER COMMENTS

Reviewer #1 (Remarks to the Author):

Manuscript #: NCOMMS-22-50075-T

Title: Lipoprotein metabolism mediates hematopoietic stem cell responses under acute anemic conditions.

Corresponding Author: Kenichi Miharada

The paper from Saito et al. analyzes the role of hematopoietic stem cells in the response to anemia. The authors analyze bone marrow HSPC populations after phenylhydrazine (PHZ) treatment. Their data show that HSCs and MEPs increase in the bone marrow, but this increase is not due to increased Epo levels in the bone marrow. Analysis of HSCs from PHZ treated mice using a new CFU-FACS assay showed an increased production of Ter119+ cells in the colonies. Gene expression analysis showed that the major changes in expression did not involve erythroid signature genes but rather showed changes in lipid metabolism. Expression of the Vldlr correlated with HSCs that had increased erythroid potential. ApoE levels increased in PHZ treated mice and ApoE treatment of HSCs increased erythroid potential. ATAC-seq analysis showed that Vldlrhi HSCs from PHZ treated mice showed increased areas of closed chromatin when compared to Vldlrlo HSCs. Motif analysis showed that the areas with increased closed chromatin were enriched in Erg1 motifs. Inhibition of Erg1 showed similar effects as ApoE treatment in increasing erythroid potential of HSCs.

Overall, this paper identifies a new component of the anemia response which has not been previously characterized. The major question remains, however, as to how much this response by HSCs contributes to the recovery from anemia? Furthermore, the conclusion that the response is independent of Epo is based on Epo levels in the bone marrow, not on a lack of Epo-signaling responses.

Specific Criticisms.

1. Could the increase in HSC proliferation observed in Figure 1 be due to inflammation? It is not clear how figures 1D and 1E are different. The legend for 1D states that it is a frequency, but the axes are labeled the same as 1E.
2. I think figure 2A is missing.
3. In the transplant experiments do any of the KuO labeled cells home to the spleen? Do any of these cells form BFU-E? Spleen derived erythrocytes could give rise to the labeled erythrocytes post transplant.
4. What day was the analysis of Epo concentrations in shown Figure 3A done?
5. I think it is difficult to make the conclusion that the decreased expression CD163 is a mechanism for HSCs to avoid free radical attacks.
6. You describe the changes in Vldl, but it looks like there were also significant changes in LDL as well. Could the authors discuss why changes in Vldl lead to changes in erythroid potential?
7. Can the authors comment that the Vldlrlo HSCs gave rise to more GMMk colonies, could these HSCs be more myeloid biased?
8. The response to acute PHZ anemia is known to mobilize stress erythroid progenitors in the spleen, how would you compare the magnitude of that response with what you observed in the bone marrow HSCs?
9. PHZ induces an inflammatory response and pro-inflammatory signals are known to inhibit bone marrow erythropoiesis. How would those signals affect the ability of HSCs to respond to anemia? Others have shown that bone marrow BFU-E decrease during the recovery from PHZ induced acute anemia. Do

you think the Vldlrhi HSCs generate erythrocytes by generating BFU-E?

10. Your data show that ApoE treatment increases the erythroid potential of HSCs, but Sanchez et al. (Scientific Reports (2020) 10:18469 | <https://doi.org/10.1038/s41598-020-74665-x>) showed that ApoE^{-/-} mice exhibit increased stress erythropoiesis in the spleen. Could the authors comment on the differences between their data and your data?

11. Could authors comment in the discussion how the changes they observed in lipid metabolism could lead to changes in chromatin structure?

Reviewer #2 (Remarks to the Author):

Saito and co-workers have explored the role of lipoprotein metabolism during acute anemia. They have found a subset of HSCs that increase the expression of the VLDL receptor in response to anemia. These VLDLr high HSCs have an altered chromatin structure, resulting in an increased number of closed transcription sites. In particular they find that the transcription factor ERG is close in these cells during acute anemia. If they give apoE they can influence HSCs down the erythroid lineage, which is also associated with closed Erg. Inhibiting Erg or administering ApoE yielded similar results. Overall, they show the importance of lipoprotein metabolism to recovery from anemia.

Comments:

1. It appears that data referred to in the text as figure 2C is missing.
2. Are the VLDLr high HSCs CD41 low following PHZ?
3. The authors clearly show plasma cholesterol metabolism is altered in figure 4. The gene expression data also suggested changes in cellular lipid metabolism. However, it should be determined if the VLDLr high HSCs have increase lipid uptake.
4. Interestingly Abca1, a cholesterol efflux gene, is also induced in these cells following PHZ (figure 4B). This could be stimulating cholesterol removal from the cell.
5. In figure 6B PHZ failed to elicit a erythroid response in the Apoe KO mice. These appear to be total body KO mice and hence will have very altered VLDL metabolism. Apoe KO HSCs have been shown previously respond in a cell intrinsic manner in producing myeloid cells. The authors should perform a bone marrow transplant to explore the role of Apoe. Moreover, a competitive BMT would be ideal.
6. Alternatively, if it is not cell intrinsic can the VLDL particles from WT mice be transferred into Apoe KO mice injected with PHZ.
7. How is the injected apoE functioning in the mice? presumably it goes into particles?
8. In the expements with the Erg +/- HSCs, what is the functional data on the downstream cells and RBCs? Is it the same at the inhibitor?
9. Minor: In the discussion they authors mention hypercholesterolemia alters HSC mobilisation – the paper from Westerterp Cell Stem Cell (2012) should be mentioned.

Reviewer #3 (Remarks to the Author):

In this study, the authors assessed and found an erythroid biased HSC expansion during acute anemia. Further analysis showed parallel up-regulation of plasma apoE levels and VLDLR expression in HSCs following PHZ induced hemolysis. Erythropoiesis ex vivo appeared to be reduced in the absence of apoE and apoE added stimulated erythropoiesis. The authors concluded that this study has identified a non-canonical pathway regulating HSC differentiation in an erythroid biased fashion and involving apoE rich VLDL lipoprotein metabolism. Overall, this conclusion of the study is largely supported by circumstantial, indirect and preliminary evidence and key and detailed mechanistic studies are lacking. Substantial additional mechanistic studies are needed to prove or disprove the conclusions.

1. The main novelty of this study is the attempted mechanistic link in apoE/VLDL-VLDLR-HSC erythroid biased expansion in acute anemia. The studies should have been focused on it to provide convincing evidence. However, this was not the case. Using ApoE^{-/-} mice, only Fig 6B is shown which showed non-significant increase in erythropoiesis from HSCs ex vivo following PHZ treatment in ApoE^{-/-} mice. In this single assay pooled from three experiments, the positive control (WT vs WT/PHZ) was lacking. As a key experiment to support a major hypothesis, this is a significant miss. What is the impact of ApoE deficiency on HSC expansion in vivo in response to PHZ like shown in Fig 1C in wild type mice? How about the impact on peripheral RBC counts and Hb during the acute induced anemia and recovery phase? All these key data are lacking.
2. The authors assumed that VLDLR upregulation in HSCs is important. This hypothesis could be assessed by assessing HSC response to PHZ in Vldlr^{-/-} mice (available from Jackson Lab). This obvious key experiment is lacking.
3. The authors found markedly decreased plasma TG in VLDL and reduced VLDL size but increased apoE in response to PHZ. The authors need to use Western analysis with HPLC fractions to determine with which lipoprotein fraction the increased apoE was associated. TG and cholesterol lipoprotein profiles from each fraction of HPLC need to be shown probably in supplemental data to better understand it.
4. During PHZ induce acute anemia, male but not female BM HSC and MEP was markedly expanded although in both male and female the recovery of peripheral RBC was comparable. The authors attributed this to more pronounced splenic HSC expansion in female. That is insufficient as the authors did not show splenic MEP expansion in male and female and it is not clear whether it is different.
5. PHZ induces hemolysis and release of Hb and heme. These cytotoxic factors could cause inflammatory responses and subsequent impact on HSC differentiation and expansion, as introduced by the authors at the beginning of the manuscript in examples such as infection. ApoE is also an important molecule involved in regulation of innate immunity. In combination with PHZ, apoE deficiency could regulate HSC expansion in unexpected ways via regulation of inflammation. This has not been considered. Phlebotomy is a better approach but only limitedly used in this study. ApoE and VLDLR deficient mice should have been used in combination with phlebotomy to assess HSC expansion and erythropoiesis. The altered lipid metabolism in HSCs should be assessed using phlebotomy models as well.
6. While plasma VLDL size and TG were markedly decreased by PHZ, ApoE but not VLDL was given in ex vivo studies to stimulate erythropoiesis. Added apoE may not provide additional TG to the cultured cells. How does apoE alter erythropoiesis? ApoE added could promote cholesterol efflux. Binding of apoE to cell surface receptor could change cell signaling. The importance of VLDLR and other apoE receptors is not known. While not necessarily conflicting with the current study, it is intriguing that recent study indicates erythropoiesis is significantly increased in apoE^{-/-} mice fed high fat diet relative to chow diet (PMID:33116141). This needs to be discussed.
7. It is not clear why the authors assessed the impact of Hp and Hpx but not Hb and heme on

erythropoiesis and megakaryopoiesis ex vivo.

Reviewer #4 (Remarks to the Author):

It has been documented that HSCs can respond to a variety of stresses. However, how HSCs respond to anemia remains largely unclear. In the present study, the authors showed that HSCs expanded upon acute anemia induction due to increased proliferation. They further showed that the erythroid differentiation potential of HSCs from anemic mice were enhanced. Notably, acute anemia induction led to significant increase in the levels of EPO in peripheral blood but not in the bone marrow fluid. Instead, acute anemia induction led to changes in the expression of lipoprotein related genes in HSCs with the expression of Vldlr significantly upregulated in HSCs, suggesting the potential role of Vldlr in acute anemia-induced changes of HSC function (enhanced erythroid proliferation and differentiation potential). This is supported by their findings that Vldlr^{high} HSC had stronger erythroid differentiation potential than Vldlr^{low} HSC and that ApoE promoted erythroid differentiation of HSCs. Finally, the authors showed that acute anemia induction and ApoE caused epigenetic changes, particularly the changes in chromatin accessibility of Erg binding motif. They conclude that acute anemia led to enhanced proliferation and erythroid differentiation potential of HSCs in an EPO-independent manner. Although some of the findings are interesting, but there are issues that need to be addressed.

1) Figure 1D shows that in male mice after PHZ injection, HSC and MEP increased and then returned to baseline level. However, surprisingly the number of Ter119⁺ cells did not change at all. This does not make sense because anemia usually leads to dramatic changes in erythropoietic activity in the bone marrow and spleen.

2) Figure 2D shows that the % of KuO⁺ RBC in PB increased significantly 2 weeks after transplantation. How about the percentage of KuO⁺ lineage cells in bone marrow or spleen? (Please note that the labels Fig 2 were not totally correct).

3) Figure 3A and 3B shows the levels of EPO, SCF, CXCL12 in plasma and bone marrow fluid respectively. In the case of EPO, it is well established that EPO is mainly produced by the kidney and circulates in the blood stream. Thus it is expected that the levels of EPO should be similar between plasma and bone marrow fluid. How the authors explain the significantly higher EPO levels in bone marrow fluid (appears to be more than 10 times) than that in plasma at steady state? Similarly, how the authors explain acute anemia induction led to than 20 times increase in EPO levels in plasma but not change in bone marrow fluid?

4) Figure 3E shows that some significantly up-regulated genes (such as Tfrc and Socs3) in HSCs of the acute anemia mice are well-known downstream genes of the EPO signal, indicating the involvement of EPO/EPOR signaling. This finding do not support the authors' conclusion that acute anemia-induced HSC expansion is EPO-independent.

5) Figure 4D shows changes in lipoprotein profiling in peripheral blood of PHZ-injected mice. Do these changes also exist in the phlebotomy mouse model?

6) Figure 4F shows that PHZ injection led to significantly increased expression of Vldlr on HSCs. It will be interesting to know the mechanisms for the increased expression of Vldlr upon acute anemia induction.

7) Figure 5, the ATAC analyses were performed on only two samples from each group. Although PCA analyses revealed separation of the samples, but the separation is not that great. For example, one PHZ-

VldlrHigh is more close to PHZ-VldlrLow than to another PHZ-VldlrHigh. In general, analyses should be performed with at least three samples.

8) Figure 6A shows very transit increase in ApoE levels in both plasma and bone marrow liquid, with increasing on day 1 following PHZ injection and returning to baseline on day 2. Fig 6C to 6I show the effects of ApoE erythroid differentiation, chromatin accessibility after treatment for 3 days. To mimic the situation in vivo, ApoE treatment should be for one day. It is unclear the authors chose to treat the cells for three days. Additionally, dose-dependent effects of ApoE should also be examined.

9) The content of the paragraph under the title "HSCs under acute anemia elevate erythroid-signature gene expression" does not seem to be consistent with the title. This paragraph mainly described various changes without mentioning erythroid-signature genes are elevated.

Re: NCOMMS-22-50075-T
Point-by-point response

We would like to thank the reviewers for careful and constructive criticism of our paper entitled “Lipoprotein metabolism mediates hematopoietic stem cell responses under acute anemic conditions”. To address the criticism and improve the paper, we have performed additional experiments and added new data. Additionally, we have reorganized the paper and corrected the text to improve the presentation. **In the revised manuscript, the new data and text are indicated in red.** New figure numbers due to re-layout are also shown in red. Below is a point-by-point response to the reviewers’ suggestions and critiques.

Reviewer #1 (Remarks to the Author):

The paper from Saito et al. analyzes the role of hematopoietic stem cells in the response to anemia. The authors analyze bone marrow HSPC populations after phenylhydrazine (PHZ) treatment. Their data show that HSCs and MEPs increase in the bone marrow, but this increase is not due to increased Epo levels in the bone marrow. Analysis of HSCs from PHZ treated mice using a new CFU-FACS assay showed an increased production of Ter119+ cells in the colonies. Gene expression analysis showed that the major changes in expression did not involve erythroid signature genes but rather showed changes in lipid metabolism. Expression of the Vldlr correlated with HSCs that had increased erythroid potential. ApoE levels increased in PHZ treated mice and ApoE treatment of HSCs increased erythroid potential. ATAC-seq analysis showed that Vldlrhi HSCs from PHZ treated mice showed increased areas of closed chromatin when compared to Vldlrlo HSCs. Motif analysis showed that the areas with increased closed chromatin were enriched in Erg1 motifs. Inhibition of Erg1 showed similar effects as ApoE treatment in increasing erythroid potential of HSCs.

Overall, this paper identifies a new component of the anemia response which has not been previously characterized. The major question remains, however, as to how much this response by HSCs contributes to the recovery from anemia? Furthermore, the conclusion that the response is independent of Epo is based on Epo levels in the bone marrow, not on a lack of Epo-signaling responses.

We would like to thank the reviewer for the constructive and valuable comments.

Specific Criticisms.

1. Could the increase in HSC proliferation observed in Figure 1 be due to inflammation? It is not clear how figures 1D and 1E are different. The legend for 1D states that it is a frequency, but the axes are labeled the same as 1E.

Thank you very much for asking the important point. We initially wondered whether inflammation was induced upon PHZ treatment; therefore, we performed GSEA on the microarray data shown in Fig. 3e. As the result, we did not see significant enrichment of inflammatory signatures in HSCs of both PHZ-treated and phlebotomized mice. The data is now shown in **Supplementary Fig. 3b**. In addition, we performed a Cytokine-Array, which comprehensively analyzes the existence of 144 cytokines, including inflammatory cytokines such as IL-6, in the bone marrow fluid. However, the assay did not observe clear induction of any cytokine. This data is now shown in **Supplementary Fig. 3c**. Thus, we think inflammation may not be a major trigger of the HSC response. We appreciate the comment on the figure legend; it has been corrected.

2. I think figure 2A is missing.

Thank you for your careful reading. We realized that the figures and their legends did not match; they have been corrected.

3. In the transplant experiments do any of the KuO labeled cells home to the spleen? Do any of these cells form BFU-E? Spleen derived erythrocytes could give rise to the labeled erythrocytes post transplant.

Unfortunately, we did not analyze the spleen of the engrafted mice. In this experiment, all transplanted HSCs were harvested from the BM of PHZ- (or PBS-) treated mice.

4. What day was the analysis of Epo concentrations in shown Figure 3A done?

These are the data from the mice 3 days post PHZ injection or phlebotomy. This information has been added to the figure legend.

5. I think it is difficult to make the conclusion that the decreased expression CD163 is a mechanism for HSCs to avoid free radical attacks.

Thank you very much for the important suggestion. We agree, our data is not sufficient to draw the conclusion. We have moderated the statement.

6. You describe the changes in Vldl, but it looks like there were also significant changes in LDL as a well. Could the authors discuss why changes in Vldl lead to changes in erythroid potential?

Since VLDL, but not LDL, contains ApoE as one of the components, degradation of VLDL is expected to be the main supply of the released ApoE. This is described in the section "ApoE controls the erythroid differentiation potential of HSCs".

7. Can the authors comment that the Vldl^{low} HSCs gave rise to more GMMk colonies, could these HSCs be more myeloid biased?

Our CFU-FACS data separately analyzing the colony forming capacity of Vldl^{high} and Vldl^{low} HSCs (Fig. 4g) did not show a significant difference in their myeloid cell contents; therefore, it is not possible to say whether Vldl^{low} HSCs are myeloid bias.

8. The response to acute PHZ anemia is known to mobilize stress erythroid progenitors in the spleen, how would you compare the magnitude of that response with what you observed in the bone marrow HSCs?

Thank you very much for raising an important discussion. It is of great interest but also challenging to demonstrate how HSCs and erythroid progenitor cells (particularly splenic cells) contribute differently to acute anemia responses (since HSCs also migrate to the spleen). What we want to propose in this study is whether and how HSCs respond to severe anemic stress.

9. PHZ induces an inflammatory response and pro-inflammatory signals are known to inhibit bone marrow erythropoiesis. How would those signals affect the ability of HSCs to respond to

anemia? Others have shown that bone marrow BFU-E decrease during the recovery from PHZ induced acute anemia. Do you think the Vldlrhi HSCs generate erythrocytes by generating BFU-E?

As commented on your question #1, at least we did not see clear signs of inflammation signals in the HSCs upon PHZ treatment. Our CFU-FACS data did not find any “BFU-E” colony according to its definition, but this is because the experiments were done using HSCs. Although we do not have direct evidence, it must be reasonable to say VLDLR^{high} HSCs generate erythrocytes through BFU-Es.

10. Your data show that ApoE treatment increases the erythroid potential of HSCs, but Sanchez et al. (Scientific Reports (2020) 10:18469 | <https://doi.org/10.1038/s41598-020-74665-x>) showed that ApoE^{-/-} mice exhibit increased stress erythropoiesis in the spleen. Could the authors comment on the differences between their data and your data?

Thank you very much for raising the important point. The main difference is that Sanchez et al. analyzed total erythropoiesis and the splenic response while we focused on the HSC response. It is not surprising that BM-HSC and spleen-migrating erythroid progenitors show different reactions.

11. Could authors comment in the discussion how the changes they observed in lipid metabolism could lead to changes in chromatin structure?

This is one of the unanswered questions in this study; considering the change in Erg binding site without alterations in its expression level, it is possible that lipid metabolism could directly modify the binding between transcription factors and their target genomic regions, which controls transcription levels. In fact, it has been proposed that small lipid molecules such as fatty acids and cholesterols can bind to chromatin and regulate gene expression (Fernandes et al., *Prog Lipid Res.*, 2018). Since we have seen the induction of Fabp5, a protein transporting fatty acid, it is unsurprising if fatty acids bind to chromatin and affect chromatin accessibility (or target Erg protein directly). **We now discuss it in the Discussion section**, but in the future, performing a chromatin structure assay such as Hi-C would be interesting (we need quite a lot of cells, though).

Reviewer #2 (Remarks to the Author):

Saito and co-workers have explored the role of lipoprotein metabolism during acute anemia. They have found a subset of HSCs that increase the expression of the VLDL receptor in response to anemia. These VLDLr high HSCs have an altered chromatin structure, resulting in an increased number of closed transcription sites. In particular they find that the transcription factor ERG is close in these cells during acute anemia. If they give apoE they can influence HSCs down the erythroid lineage, which is also associated with closed Erg. Inhibiting Erg or administering ApoE yielded similar results. Overall, they show the importance of lipoprotein metabolism to recovery from anemia.

We would like to thank the reviewer for the constructive and valuable comments.

Comments:

1. It appears that data referred to in the text as figure 2C is missing.

Thank you very much for your careful reading. It has been corrected.

2. Are the VLDLr high HSCs CD41 low following PHZ?

We analyzed the expression of CD41 on both Vldlr^{high} and Vldlr^{low} HSCs before and after PHZ treatment. The assay revealed that CD41 expression showed a transient and modest increase on Vldlr^{high} HSCs at day 1 after PHZ treatment.

However, as shown in **Supplementary Fig. 2c**, Vldlr^{high} and Vldlr^{low} HSCs at 1-day after PHZ treatment did not show significant differences in the lineage distribution in their colonies, indicating that the CD41 induction does not have much impact on the cellular potential.

3. The authors clearly show plasma cholesterol metabolism is altered in figure 4. The gene expression data also suggested changes in cellular lipid metabolism. However, it should be determined if the VLDLr high HSCs have increase lipid uptake.

Thank you very much for the very interesting input. We measured fatty acid uptake using BODIPY. However, we did not see significant difference (Please see below). More sensitive assays may be needed. However, such assays are usually challenging for HSC studies since they require a lot of cells.

4. Interestingly *Abca1*, a cholesterol efflux gene, is also induced in these cells following PHZ (figure 4B). This could be stimulating cholesterol removal from the cell.

We appreciate this suggestion which is correlated to the comment above (#3). We performed cellular cholesterol staining using Filipin III staining. Indeed, we could see that Filipin III intensity was lower in the HSCs after PHZ treatment than the control HSCs, although the difference was insignificant, supporting the finding of *Abca1* induction. The lower cholesterol content was also seen in GMPs.

5. In figure 6B PHZ failed to elicit an erythroid response in the *ApoE* KO mice. These appear to be total body KO mice and hence will have very altered VLDL metabolism. *ApoE* KO HSCs have been shown previously respond in a cell intrinsic manner in producing myeloid cells. The authors should perform a bone marrow transplant to explore the role of *ApoE*. Moreover, a competitive BMT would be ideal.

Thank you very much for indicating this important point. We totally agree that *ApoE* is highly expressed in HSCs, and the data shown in Fig. 6b used *ApoE* total KO mice. To exclude the possible impact of endogenous *ApoE* depletion, as suggested, we first transplanted wild-type (WT) bone marrow (BM) to WT mice or *ApoE* KO mice. Three months later, we then injected these engrafted mice with PHZ and performed CFU-FACS by sorting donor-derived HSCs. This experiment showed very similar data with the previous setting: no increase in the erythroid content in HSCs engrafted to *ApoE* KO mice upon PHZ treatment, while HSCs engrafted to WT mice showed a significant increase in the erythroid content. This experiment confirmed that lack of the response against acute anemia induction in *ApoE* depletion was due to the loss of environmental ApoE but not the endogenous *ApoE* expression in HSCs. This data is now shown in new Fig. 6c and 6d.

6. Alternatively, if it is not cell intrinsic can the VLDL particles from WT mice be transferred into *ApoE* KO mice injected with PHZ.

This experiment sounds interesting but more complicated, as purification of VLDL with their normal activity is quite challenging. We therefore selected the approach above.

7. How is the injected apoE functioning in the mice? presumably it goes into particles?

We didn't inject ApoE into mice. All of the experiments using recombinant ApoE are *in vitro* cultures of HSCs.

8. In the experiments with the *Erg* +/- HSCs, what is the functional data on the downstream cells and RBCs? Is it the same at the inhibitor?

The information regarding *Erg*^{+/-} mice is based on literature; we have not used *Erg*-deficient mice. It would however be interesting to analyze *Erg*-deficient from the aspect of stress erythropoiesis.

9. Minor: In the discussion they authors mention hypercholesterolemia alters HSC mobilisation – the paper from Westerterp Cell Stem Cell (2012) should be mentioned.

Thank you very much for introducing the important paper. It is now cited and included as a reference.

Reviewer #3 (Remarks to the Author):

In this study, the authors assessed and found an erythroid biased HSC expansion during acute anemia. Further analysis showed parallel up-regulation of plasma apoE levels and VLDLR expression in HSCs following PHZ induced hemolysis. Erythropoiesis ex vivo appeared to be reduced in the absence of apoE and apoE added stimulated erythropoiesis. The authors concluded that this study has identified a non-canonical pathway regulating HSC differentiation in an erythroid biased fashion and involving apoE rich VLDL lipoprotein metabolism. Overall, this conclusion of the study is largely supported by circumstantial, indirect and preliminary evidence and key and detailed mechanistic studies are lacking. Substantial additional mechanistic studies are needed to prove or disprove the conclusions.

We would like to thank the reviewer for the constructive and valuable comments. We have added new data to address the criticism.

1. The main novelty of this study is the attempted mechanistic link in apoE/VLDL-VLDLR-HSC erythroid biased expansion in acute anemia. The studies should have been focused on it to provide convincing evidence. However, this was not the case. Using *ApoE*^{-/-} mice, only Fig 6B is shown which showed non-significant increase in erythropoiesis from HSCs ex vivo following PHZ treatment in *ApoE*^{-/-} mice. In this single assay pooled from three experiments, the positive control (WT vs WT/PHZ) was lacking. As a key experiment to support a major hypothesis, this is a significant miss. What is the impact of *ApoE* deficiency on HSC expansion in vivo in response to PHZ like shown in Fig 1C in wild type mice? How about the impact on peripheral RBC counts and Hb during the acute induced anemia and recovery phase? All these key data are lacking.

Thank you very much for the important criticism. First, we also considered that the previous experiment was not convincing; one of the reasons was to utilize conventional *ApoE* total KO mice even though *ApoE* is expressed in HSCs (Supplementary Fig. 6a). As a better approach, therefore, we first transplanted wild-type (WT) bone marrow (BM) to WT mice or *ApoE* KO mice. Three months later, we then injected these engrafted mice with PHZ and performed CFU-FACS by sorting donor-derived HSCs. This experiment showed very similar data with the previous setting: no increase in the erythroid content in HSCs engrafted to *ApoE* KO mice upon PHZ treatment, while HSCs engrafted to WT mice showed a significant increase in the erythroid content. This experiment confirmed that lack of the response against acute anemia induction in *ApoE* depletion was due to the loss of environmental ApoE but not the endogenous *ApoE* expression in HSCs. This data is now shown in new Fig. 6c and 6d. While we see alleviated differentiation shift of HSCs in *ApoE* KO mice, expansion of the HSC fraction was still seen although the difference was insignificant. Thus it is inconclusive but ApoE might not be involved in the expansion of HSCs upon acute anemia induction.

2. The authors assumed that VLDLR upregulation in HSCs is important. This hypothesis could be assessed by assessing HSC response to PHZ in *Vldlr*^{-/-} mice (available from Jackson Lab). This obvious key experiment is lacking.

Thank you very much for the nice suggestion. However, *Vldlr* deficient mice are smaller and lean, as their altered lipid metabolism affects the whole body (Frykman et al., *Proc Natl Acad Sci USA*, 1995). Thus, it would be hard to evaluate what actually impacted the HSCs. Instead, we utilized validated shRNA available from Sigma-Aldrich to knock down the expression of *Vldlr*. We sorted HSC from WT mice and infected them with the lentivirus expressing shRNA targeting *Vldlr* or luciferase (control). Then, the cells were cultured with or without recombinant ApoE for three days, and CFU-FACS was performed. As a result, we observed that *Vldlr*-knockdown alleviated the increase in erythroid potential upon the addition of ApoE protein, while shluciferase control cells still showed higher erythroid content. This data supports the fact that *Vldlr* in HSC plays a major role in the HSC's response to ApoE. The data is now shown in **new Supplementary Fig. 6b**.

3. The authors found markedly decreased plasma TG in VLDL and reduced VLDL size but increased apoE in response to PHZ. The authors need to use Western analysis with HPLC fractions to determine with which lipoprotein fraction the increased apoE was associated. TG and cholesterol lipoprotein profiles from each fraction of HPLC need to be shown probably in supplemental data to better understand it.

In the lipoprotein analysis, all fractions were separated and immediately analyzed by a mass spectrometer. I have discussed this with the company, but unfortunately, it is difficult to collect each fraction and use it for Western blotting analysis.

4. During PHZ induce acute anemia, male but not female BM HSC and MEP was markedly expanded although in both male and female the recovery of peripheral RBC was comparable. The authors attributed this to more pronounced splenic HSC expansion in female. That is insufficient as the authors did not show splenic MEP expansion in male and female and it is not clear whether it is different.

Thank you very much for pointing out the important issue. We agree with the reviewer, our data was insufficient to conclude as written in the previous version of the text. We analyzed the spleen of both male and female mice after PHZ injection. Unexpectedly, we did not see a better expansion MEP; however, we saw significantly higher increase in Ter119⁺ erythroid progenitor cells in the female spleen. This data is now shown in **Supplementary Fig. 1d and 1e**, and the corresponding text has been revised.

5. PHZ induces hemolysis and release of Hb and heme. These cytotoxic factors could cause inflammatory responses and subsequent impact on HSC differentiation and expansion, as introduced by the authors at the beginning of the manuscript in examples such as infection. ApoE is also an important molecule involved in regulation of innate immunity. In combination with PHZ, apoE deficiency could regulate HSC expansion in unexpected ways via regulation of inflammation. This has not been considered. Phlebotomy is a better approach but only limitedly used in this study. ApoE and VLDLR deficient mice should have been used in combination with phlebotomy to assess HSC expansion and erythropoiesis. The altered lipid metabolism in HSCs should be assessed using phlebotomy models as well.

We checked if the inflammation signature is elevated in the HSC of PHZ- or phlebotomized mice by performing GSEA on the microarray data shown in Fig. 3e. We did not see significant enrichment of inflammatory signatures in both HSCs of PHZ- and phlebotomized mice. The data is now shown in Supplementary Fig. 3b. Thus, we think inflammation signal is not triggered by PHZ/phlebotomy. This is probably due to the lowered CD163 expression which mediates the scavenging system and potentially transmits the heme-oriented inflammation signal to HSCs.

6. While plasma VLDL size and TG were markedly decreased by PHZ, ApoE but not VLDL was given in ex vivo studies to stimulate erythropoiesis. Added apoE may not provide additional TG to the cultured cells. How does apoE alter erythropoiesis? ApoE added could promote cholesterol efflux. Binding of apoE to cell surface receptor could change cell signaling. The importance of VLDLR and other apoE receptors is not known. While not necessarily conflicting with the current study, it is intriguing that recent study indicates erythropoiesis is significantly increased in *apoe*^{-/-} mice fed high fat diet relative to chow diet (PMID:33116141). This needs to be discussed.

Thank you very much for raising the important topic. We consider that the chromatin accessibility change (lowering accessibility of HSC/megakaryocyte-related genes) upon ApoE binding to Vldlr is the key mechanism of the HSC responses to acute anemia induction as described in the discussion part; however, it still remains unclear how ApoE changes the chromatin accessibility. This is of great interest, but so far we have not found the exact mechanism and further study is necessary. Regarding the cholesterol efflux, which was also suggested by Reviewer #2, we performed cellular cholesterol staining using Filipin III. We could see that Filipin III intensity was lower in the HSCs after PHZ treatment than the control HSCs, although the difference was insignificant (please see below). Thus, as the reviewers indicate this lowered cellular cholesterol may be involved in the ApoE-oriented HSC response.

The *ApoE* KO mouse study should be cited and discussed, even though the setting of the study is different from our study. The paper is now cited and discussed in the text.

7. It is not clear why the authors assessed the impact of Hp and Hpx but not Hb and heme on erythropoiesis and megakaryopoiesis ex vivo.

We focused on Hp and Hpx since the microarray data of day 3 HSCs showed a reduction of CD163 and CD91 (Fig. 3e), which are receptors for Hp and Hpx, respectively. Hb and Heme are captured by Hp and Hpx, and then taken up by CD163 and CD91. However, we understand the reviewer's concern about the potential direct effect of Hb and Heme not through CD163/CD91. We, therefore, performed CFU-FACS of HSCs treated with hemin. In addition, SnPP (heme oxygenase inhibitor) is known to ameliorate anemia in the thalassemia model (Garcia-Santos et al., Blood, 2018), so we also tested SnPP. We observed no difference by adding hemin or SnPP (please see below); we did not test a direct addition of Hb itself, however,

it is uneasy and would be highly toxic. Thus, we conclude that scavenging signal probably affects HSCs through CD163 or CD91.

Reviewer #4 (Remarks to the Author):

It has been documented that HSCs can respond to a variety of stresses. However, how HSCs respond to anemia remains largely unclear. In the present study, the authors showed that HSCs expanded upon acute anemia induction due to increased proliferation. They further showed that the erythroid differentiation potential of HSCs from anemic mice were enhanced. Notably, acute anemia induction led to significant increase in the levels of EPO in peripheral blood but not in the bone marrow fluid. Instead, acute anemia induction led to changes in the expression of lipoprotein related genes in HSCs with the expression of Vldlr significantly upregulated in HSCs, suggesting the potential role of Vldlr in acute anemia-induced changes of HSC function (enhanced erythroid proliferation and differentiation potential). This is supported by their findings that Vldlr^{high} HSC had stronger erythroid differentiation potential than Vldlr^{low} HSC and that ApoE promoted erythroid differentiation of HSCs. Finally, the authors showed that acute anemia induction and ApoE caused epigenetic changes, particularly the changes in chromatin accessibility of Erg binding motif. They conclude that acute anemia led to enhanced proliferation and erythroid differentiation potential of HSCs in an EPO-independent manner. Although some of the findings are interesting, but there are issues that need to be addressed.

We would like to thank the reviewer for the constructive and valuable comments. We have added new data to address the criticism.

1) Figure 1D shows that in male mice after PHZ injection, HSC and MEP increased and then returned to baseline level. However, surprisingly the number of Ter119⁺ cells did not change at all. This does not make sense because anemia usually leads to dramatic changes in erythropoietic activity in the bone marrow and spleen.

Thank you very much for pointing out it. we now include the data of the spleen, showing a significant increase in Ter119⁺ erythroid progenitor cells which was more pronounced in the female spleen. This data is now shown in **Supplementary Fig. 1e**,

2) Figure 2D shows that the % of KuO⁺ RBC in PB increased significantly 2 weeks after transplantation. How about the percentage of KuO⁺ lineage cells in bone marrow or spleen? (Please note that the labels Fig 2 were not totally correct).

We monitored the PB of the transplanted mice over 3 months and sacrificed them after the experimental period; therefore, unfortunately, we do not have BM and spleen data at 2 weeks post transplantation.

3) Figure 3A and 3B shows the levels of EPO, SCF, CXCL12 in plasma and bone marrow fluid respectively. In the case of EPO, it is well established that EPO is mainly produced by the kidney and circulates in the blood stream. Thus it is expected that the levels of EPO should be similar between plasma and bone marrow fluid. How the authors explain the significantly higher EPO levels in bone marrow fluid (appears to be more than 10 times) than that in plasma at steady state? Similarly, how the authors explain acute anemia induction led to than 20 times increase in EPO levels in plasma but not change in bone marrow fluid?

This was also surprising to us. So far, unfortunately, we do not have a convincing resource to answer this interesting question. First, we want to clarify that the PB plasma and BM fluid collection methods are different, so we cannot directly compare EPO concentrations between these two sources. We can just speculate why only in PB plasma EPO concentration increases;

EPO is normally produced in kidney and secreted to the blood circulation. The cells that require and consume EPO, being erythroblasts, mainly exist in BM and spleen. Other cytokines could be produced in liver and BM cells as well (it has been reported that concentrations of many cytokines, particularly inflammatory cytokines, are similar between PB and BM under physiological conditions). Therefore, EPO concentration presumably easier increases in the PB than in BM. In contrast, we consider that ApoE is released from VLDL degradation; VLDL exist also in the BM. These are of great interest to answer, however, we need more thorough study.

4) Figure 3E shows that some significantly up-regulated genes (such as *Tfrc* and *Socs3*) in HSCs of the acute anemia mice are well-known downstream genes of the EPO signal, indicating the involvement of EPO/EPOR signaling. This finding do not support the authors' conclusion that acute anemia-induced HSC expansion is EPO-independent.

As the reviewer point out, these genes are known as downstream of EPO/EPOR signaling. To test whether EPO/EPOR actually stimulate HSCs, we performed intracellular staining of pSTAT3/pSTAT5, which must increase upon EPO/EPOR. Our flow cytometry analyses did not detect any change in both pSTAT3 and pSTAT5 (please see below). Furthermore, GSEA using our microarray data observed no-enrichment of EPO signals. *Tfrc* expression is known to be induced by EPO-independent signals such as hypoxia and inflammation (Fagundes et al., *Front Physiol.*, 2022). Additionally, *Socs3* is induced not only by EPO but also other cytokine signals. Thus, EPO-independent signal is involved in the induction of *Tfrc/Socs3* genes.

5) Figure 4D shows changes in lipoprotein profiling in peripheral blood of PHZ-injected mice. Do these changes also exist in the phlebotomy mouse model?

This is a quite reasonable comment and we also wanted to do, however, the assay requires a relatively large volume of samples and it was challenging to collect sufficient volume of blood plasma after repeated phlebotomy. Therefore we cannot provide the data.

6) Figure 4F shows that PHZ injection led to significantly increased expression of *Vldlr* on HSCs. It will be interesting to know the mechanisms for the increased expression of *Vldlr* upon acute anemia induction.

Thank you for asking this interesting question. According to literatures, unfolded protein stress (UPR, also known as ER stress) is known to induce *Vldlr* expression in murine hepatocytes (Jo et al., *Hepatology*, 2013). Since GSEA of our microarray data indicated the induced UPR signal both upon PHZ treatment and phlebotomy (please see below), it is possible that the upregulation of *Vldlr* expression reflects UPR induction.

7) Figure 5, the ATAC analyses were performed on only two samples from each group. Although PCA analyses revealed separation of the samples, but the separation is not that great. For example, one PHZ-VldlrHigh is more close to PHZ-VldlrLow than to another PHZ-VldlrHigh. In general, analyses should be performed with at least three samples.

Many papers published in Nature Communications and other major journals show ATAC-seq data with two samples, so we consider this to be fine. This is usually accepted if the samples are rare, such as HSCs. In addition, we performed motif analysis separately, and both data sets found Erg as the most enriched motif, suggesting that these data are similar.

Examples of papers performing ATAC-seq with 2 samples:

<https://www.nature.com/articles/s41467-022-30440-2>

<https://www.sciencedirect.com/science/article/pii/S2213671115002763?via%3Dihub>

<https://www.nature.com/articles/s41590-018-0176-1>

8) Figure 6A shows very transit increase in ApoE levels in both plasma and bone marrow liquid, with increasing on day 1 following PHZ injection and returning to baseline on day 2. Fig 6C to 6I show the effects of ApoE erythroid differentiation, chromatin accessibility after treatment for 3 days. To mimic the situation in vivo, ApoE treatment should be for one day. It is unclear the authors chose to treat the cells for three days. Additionally, dose-dependent effects of ApoE should also be examined.

Thank you very much for the important opinion. We performed CFU-FACS right after the 1-day culture but did not see a significant difference. We also performed a dose-dependent experiment. The lower concentration did not show a clear effect, indicating that a certain level of the signal is needed for the HSC response. These data are now shown in **Supplementary Fig. 2c** and new **Fig. 6f**.

9) The content of the paragraph under the title “HSCs under acute anemia elevate erythroid-signature gene expression” does not seem to be consistent with the title. This paragraph mainly described various changes without mentioning erythroid-signature genes are elevated.

Thank you very much for the suggestion. We have changed it.

REVIEWER COMMENTS

Reviewer #1 (Remarks to the Author):

The authors have addressed my experimental concerns. In the discussion, the authors might want to address the differences between male and female mice and the possible roles of this Lipoprotein mediated erythropoiesis and BMP4 dependent stress erythropoiesis that occurs in the spleen.

Reviewer #2 (Remarks to the Author):

The authors have addressed my questions

Reviewer #3 (Remarks to the Author):

The authors have performed some additional studies to address the raised questions and concerns. There are still remaining concerns as to whether the observations and results, obtained primarily from studies using ex vivo CFU assays, are truly relevant and important in vivo, as raised in the review. It is unclear, e.g., whether some of the key physiological outcomes such as recovery of RBC counts following PHZ are altered in the genetically modified mouse models (ApoE^{-/-} or Vldlr knock-down or knock-out). A published study showed that RBC counts were increased in ApoE^{-/-} vs WT mice when fed a Western-type diet and RBC counts following PHZ treatment recovered more quickly in ApoE^{-/-} mice fed WD (PMID:33116141. BTW, this study needs to be cited and discussed). Even though there are some key differences in these studies such as the focus of the cited study on Western-type diet, conceptually these observations do not necessarily support the conclusions and concepts proposed in this study. The conclusions and concepts in this study would be more pronouncedly strengthened if the physiological outcome such as the recovery of RBCs are functionally in the same direction as proposed by the authors in the genetically modified models in vivo.

Reviewer #4 (Remarks to the Author):

The manuscript has improved. This reviewer still have following concerns:

- 1) Fig 1d, the authors used Ter119⁺ cells to reflect the changes in erythroid lineage and found no changes at all with PHZ treatment. This result does not make sense because PHZ treatment induces anemia and anemia in turn leads to increased erythropoiesis to compensate for the anemia.
- 2) The authors claim that PHZ treatment did not cause inflammation. But supplementary Fig 1a shows that WBCs were increased with PHZ treatment.
- 3) In the response letter, the authors show that PHZ treatment did not cause changes in Stat5 phosphorylation. It is not clear how the experiment was performed. It is important to show

positive control. Furthermore, the authors claim that EPO-independent signal is involved in the induction of Trfc/Socs3. Based on the authors' own data, can the authors clarify which signal pathway is involved?

4) The authors show that PHZ treatment led to both increased proliferation and enhanced erythroid differentiation potential of HSC. They conclude that the enhanced erythroid differentiation potential is due to increased expression Vldlr on HSC and increased production of Apoe by HSC. What is the mechanism for the increased proliferation?

Re: NCOMMS-22-50075A

Point-by-point response

We would like to thank the reviewers again for the careful and constructive comments on our paper entitled “Lipoprotein metabolism mediates hematopoietic stem cell responses under acute anemic conditions”. We have performed an additional experiment and modified the text to improve the presentation. **In the revised manuscript, the new data and text are indicated in red.** Below is a point-by-point response to the reviewers’ suggestions and critiques.

Reviewer #1 (Remarks to the Author):

The authors have addressed my experimental concerns. In the discussion, the authors might want to address the differences between male and female mice and the possible roles of this Lipoprotein mediated erythropoiesis and BMP4 dependent stress erythropoiesis that occurs in the spleen.

We would like to thank the reviewer for the valuable comment. Indeed, BMP4 signaling is a key player in the splenic stress erythropoiesis. We find it interesting that BMP signaling, for instance, BMPR-II signaling, is affected by sex (Mair et al., *Am J Respir Crit Care Med.*, 2015), although this is a study for pulmonary systems. In addition, BMP4 cross-talks with estrogen signals (Qian et al., *EBioMedicine*, 2016). These observations indicate a potential mechanism explaining why male and female spleen differentially responded to stress erythropoiesis. This discussion is now added to the Discussion section.

Reviewer #2 (Remarks to the Author):

The authors have addressed my questions

We would like to thank the reviewer for the warm comment.

Reviewer #3 (Remarks to the Author):

The authors have performed some additional studies to address the raised questions and concerns. There are still remaining concerns as to whether the observations and results, obtained primarily from studies using ex vivo CFU assays, are truly relevant and

important *in vivo*, as raised in the review. It is unclear, e.g., whether some of the key physiological outcomes such as recovery of RBC counts following PHZ are altered in the genetically modified mouse models (Apo $e^{-/-}$ or Vldlr knock-down or knock-out). A published study showed that RBC counts were increased in Apo $e^{-/-}$ vs WT mice when fed a Western-type diet and RBC counts following PHZ treatment recovered more quickly in Apo $e^{-/-}$ mice fed WD (PMID:33116141. BTW, this study needs to be cited and discussed). Even though there are some key differences in these studies such as the focus of the cited study on Western-type diet, conceptually these observations do not necessarily support the conclusions and concepts proposed in this study. The conclusions and concepts in this study would be more pronouncedly strengthened if the physiological outcome such as the recovery of RBCs are functionally in the same direction as proposed by the authors in the genetically modified models *in vivo*.

We would like to thank the reviewer for the valuable comment. This is a challenging assay since red blood cells are not only generated from HSCs directly but also differentiated/matured from their immediate progenitors and precursors. Thus, it is hard to evaluate how much of them are affected by the altered HSC regulation under stress erythroid conditions. However, we totally agree with the reviewer that it is important to analyze whether and how the impaired ApoE signal affects erythroid generation *in vivo*. Therefore, we injected WT or ApoE KO mice (fed with normal diet) with PHZ and analyzed their PB 7 days post PHZ treatment. As a result, blood profiling did not find significant differences; however, we see trends that RBC profiles (the RBC number, HCT, HGB) are lower in PB of ApoE KO (please see below).

As mentioned above, impaired erythroid commitment/differentiation in ApoE KO HSCs was presumably compensated or masked by the generation of RBCs from their immediate progenitors/precursors. In addition, we want to notify that ApoE conventional KO mice are larger than WT mice, and therefore, their BM cellularity is higher than that of WT. It might also affect the result.

Next, we analyzed PB, BM, and spleen of PHZ-injected WT or ApoE KO mice using flow cytometer. Here, we saw a clear trend of lower erythroid cells (Ery) in BM and spleen (Spl, please see below). This also supports the active *in vivo* role of ApoE in stress erythropoiesis.

Reviewer #4 (Remarks to the Author):

The manuscript has improved. This reviewer still have following concerns:

1) Fig 1d, the authors used Ter119+ cells to reflect the changes in erythroid lineage and found no changes at all with PHZ treatment. This result does not make sense because PHZ treatment induces anemia and anemia in turn leads to increased erythropoiesis to compensate for the anemia.

We would like to thank the reviewer for the valuable comments. We were also surprised by the relatively small increase in Ter119+ cells; however, we see a clear trend of the increase. One possible reason is that we treat BM and spleen cells with hemolytic reagent before staining; it might be the reason of the underestimated erythroid cell numbers.

2) The authors claim that PHZ treatment did not cause inflammation. But supplementary Fig 1a shows that WBCs were increased with PHZ treatment.

These cell counters could also count reticulocytes as WBC. Any of our data does not support the evidence of inflammation induction.

3) In the response letter, the authors show that PHZ treatment did not cause changes in Stat5 phosphorylation. It is not clear how the experiment was performed. It is important to show positive control. Furthermore, the authors claim that EPO-independent signal is involved in the induction of Trfc/Socs3. Based on the authors' own data, can the authors clarify which signal pathway is involved?

In this experiment, we harvested bone marrow cells from wildtype. Isolated cells were first pre-incubated in SFEM medium without any cytokine for 2 hours at 37°C, and subsequently cells were transferred to fresh SFEM containing either hEPO (3 U/mL) or mIL-6 (10 ng/mL) or in a combination of EPO and IL6 for 1 hour. Then the cells were stained for cell surface markers and fixed with BD Cytotfix/Cytoperm™ Fixation/Permeabilization Kit. The fixed/permeabilized cells were stained with anti-pSTAT5 antibody and analyzed on flow cytometer. As a positive control, we used total bone marrow cells stimulated with EPO or IL-6 (please see below).

Although Tfr and Socs3 are well known down-stream genes of the EPO signal, they are not exclusive targets of EPO. For instance, Socs3 has been suggested to be up-regulated by obesity, pregnancy and refeeding through leptin receptor (Pedroso, J.A., Ramos-Lobo, A.M. & Donato, J. SOCS3 as a future target to treat metabolic disorders.

Hormones 18, 127–136, 2019). In this manuscript we have not identified exact cytokine(s) and their signals inducing these genes, it would be an important and interesting future perspectives. We have added this point in the discussion.

Gray: No cytokine, Blue: IL-6, Red: EPO

4) The authors show that PHZ treatment led to both increased proliferation and enhanced erythroid differentiation potential of HSC. They conclude that the enhanced erythroid differentiation potential is due to increased expression Vldlr on HSC and increased production of Apoe by HSC. What is the mechanism for the increased proliferation?

Thank you very much for asking the important question. This is also of great interest to us. Unfortunately, we do not have a clear answer. One possibility is that in our microarray data, we detected induction of estrogen signals (GSEA). It has been reported that estrogen signals promote HSC cycling (Nakada et al., *Nature*, 2014). However, we used male mice in the assay. We actually analyzed blood plasma to see if estrogen or other sex hormones are induced in the blood of male mice upon stress erythropoiesis, but none of the sex hormones were changed upon PHZ injection, even in female mice (unpublished, confidential data). It would be great progress if we could identify unknown hormone(s) or other metabolites that are secreted upon stress erythropoiesis and can stimulate estrogen receptors. However, it is beyond the scope of this manuscript.

REVIEWER COMMENTS

Reviewer #3 (Remarks to the Author):

The authors have performed additional studies using PHZ induced acute anemia in WT vs Apoe^{-/-} mice and evaluated the recovery of RBCs as well as RBC precursors in bone marrow and spleen. The results appear to support their hypothesis and conclusions. It is not clear why the authors choose not to include these additional data in the manuscript. These new data should have been added and discussed, since these results were obtained from in vivo studies and complimentary to the results from ex vivo studies.

Reviewer #4 (Remarks to the Author):

This reviewer does not think the authors addressed following concerns:

1) Fig 1d, the authors used Ter119⁺ cells to reflect the changes in erythroid lineage and found no changes at all with PHZ treatment. This result does not make sense because PHZ treatment induces anemia and anemia in turn leads to increased erythropoiesis to compensate for the anemia.

“One possible reason is that we treat BM and spleen cells with hemolytic reagent before staining; it might be the reason of the underestimated erythroid cell numbers.”

This reviewer does not agree with this reasoning. Instead of Ter119⁺ cells, nucleated erythroid cells should be used to reflect the changes in erythropoiesis.

2) The authors claim that PHZ treatment did not cause inflammation. But supplementary Fig 1a shows that WBCs were increased with PHZ treatment.

“These cell counters could also count reticulocytes as WBC”.

This raises the question of the reliability of the results. Efforts should be made to distinguish reticulocytes and WBC.

3) In the response letter, the authors show that PHZ treatment did not cause changes in Stat5 phosphorylation. It is not clear how the experiment was performed. It is important to show positive control. Furthermore, the authors claim that EPO-independent signal is involved in the induction of Trfc/Socs3. Based on the authors' own data, can the authors clarify which signal pathway is involved?

“Although Trfc and Socs3 are well known down-stream genes of the EPO signal, they are not exclusive targets of EPO. For instance, Socs3 has been suggested to be upregulated by obesity, pregnancy and refeeding through leptin receptor (Pedroso, J.A., Ramos-Lobo, A.M. & Donato, J. SOCS3 as a future target to treat metabolic disorders. *Hormones* 18, 127–136, 2019).”

Can the authors give an example by which Trfc is upregulated.

Re: NCOMMS-22-50075B

Point-by-point response

We would like to thank the reviewers again for the careful and constructive comments on our paper entitled “Lipoprotein metabolism mediates hematopoietic stem cell responses under acute anemic conditions”. We have performed an additional experiment and modified the text to improve the presentation. In the revised manuscript, the new data and text are indicated in red.

REVIEWER COMMENTS

Reviewer #3 (Remarks to the Author):

The authors have performed additional studies using PHZ induced acute anemia in WT vs Apoe^{-/-} mice and evaluated the recovery of RBCs as well as RBC precursors in bone marrow and spleen. The results appear to support their hypothesis and conclusions. It is not clear why the authors choose not to include these additional data in the manuscript. These new data should have been added and discussed, since these results were obtained from in vivo studies and complimentary to the results from ex vivo studies.

Thank you very much for your kind suggestion. The data is now shown in Figure 6e and f.

Reviewer #4 (Remarks to the Author):

This reviewer does not think the authors addressed following concerns:

1) *Fig 1d, the authors used Ter119⁺ cells to reflect the changes in erythroid lineage and found no changes at all with PHZ treatment. This result does not make sense because PHZ treatment induces anemia and anemia in turn leads to increased erythropoiesis to compensate for the anemia.*

“One possible reason is that we treat BM and spleen cells with hemolytic reagent before staining; it might be the reason of the underestimated erythroid cell numbers.”

This reviewer does not agree with this reasoning. Instead of Ter119+ cells, nucleated erythroid cells should be used to reflect the changes in erythropoiesis.

It is a common understanding that erythroid cells increase upon acute anemia, as reported by many groups, including our previous works. In this paper, we never state that they do not increase; it is just based on the difference in analytical methods. Since the key issue of this paper is whether hematopoietic stem cells increase or not, this does not add additional value and confirm our findings in HSCs.

2) *The authors claim that PHZ treatment did not cause inflammation. But supplementary Fig 1a shows that WBCs were increased with PHZ treatment.*

“These cell counters could also count reticulocytes as WBC”.

This raises the question of the reliability of the results. Efforts should be made to distinguish reticulocytes and WBC.

We show two data indicating inflammation is not induced; we have shown that HSCs do not induce inflammation-related gene signature (Supplementary Figure 3b) and no inflammation-related cytokines increase upon the acute anemia induction (Supplementary Figure 3c). Even if some WBC induction happens, our data do not indicate severe inflammation occurs; and at least HSCs are not affected by the inflammation.

3) *In the response letter, the authors show that PHZ treatment did not cause changes in Stat5 phosphorylation. It is not clear how the experiment was performed. It is important to show positive control. Furthermore, the authors claim that EPO-independent signal is involved in the induction of Trfc/Socs3. Based on the authors' own data, can the authors clarify which signal pathway is involved?*

“Although Tfr and Socs3 are well known down-stream genes of the EPO signal, they are not exclusive targets of EPO. For instance, Socs3 has been suggested to be upregulated by obesity, pregnancy and refeeding through leptin receptor (Pedroso, J.A., Ramos-Lobo, A.M. & Donato, J. SOCS3 as a future target to treat metabolic disorders. Hormones 18, 127–136, 2019).”

Can the authors give an example by which *Tfrc* is upregulated.

Tfrc is absolutely a representative downstream target of EPO; however, it is not exclusive. It has been reported that *Tfrc* is induced by iron-deficiency and hypoxia.

Fagundes, R.R. et al. HIF1alpha-dependent induction of *Tfrc* by a combination of intestinal inflammation and systemic iron deficiency in inflammatory bowel disease. *Front Physiol.* 13, 889091 (2022).

Ozgür, B. et al. Hypoxia increases expression of selected blood-brain barrier transporters GLUT-1, P-gp, SLC7A5 and TFRC, while maintaining barrier integrity, in brain capillary endothelial monolayers. *Fluids Barriers CNS* 19, 1 (2022).

REVIEWERS' COMMENTS

Reviewer #1 (Remarks to the Author):

The authors have responded to the reviewer's comments. One point, however, is line 126 and 127 explaining why Ter119+ cells did not increase in the bone marrow. Previous work from Lenox et al. (Blood 2005) showed that BFU-E decrease in the bone marrow after PHZ treatment. Furthermore, PHZ induces pro-inflammatory signals which decrease erythroid terminal differentiation. You state that you did not observe an inflammatory signature but that measurement was at 3 days. Pro-inflammatory signals peak early within the first 24 hours, so you might have missed them. The lack of an observed increase in Ter119+ cells in the bone marrow is not unexpected.

Reviewer #3 (Remarks to the Author):

I have no further questions.

Re: NCOMMS-22-50075C

Point-by-point response

We would like to thank the reviewers again for the careful and constructive comments on our paper entitled “Lipoprotein metabolism mediates hematopoietic stem cell responses under acute anemic conditions”. We have modified the text to improve the presentation.

REVIEWER COMMENTS

Reviewer #1 (Remarks to the Author):

The authors have responded to the reviewer's comments. One point, however, is line 126 and 127 explaining why Ter119⁺ cells did not increase in the bone marrow. Previous work from Lenox et al. (Blood 2005) showed that BFU-E decrease in the bone marrow after PHZ treatment. Furthermore, PHZ induces pro-inflammatory signals which decrease erythroid terminal differentiation. You state that you did not observe an inflammatory signature but that measurement was at 3 days. Pro-inflammatory signals peak early within the first 24 hours, so you might have missed them. The lack of an observed increase in Ter119⁺ cells in the bone marrow is not unexpected.

Thank you very much for the precious comment. I agree that no increase in the Ter119⁺ cells is *not* unexpected from several aspects. We have deleted the sentence since it is not conclusive and is based on pure speculation. Regarding the inflammation, as you commented, the inflammation could happen immediately. Therefore, we re-performed GSEA using the Day1 microarray data, however, we did not see significant enrichment of the inflammation signature (please see the data below). This is also informative, so the data is now included in Supplementary Figure S2b. Thank you very much again for your constructive comments.

NES	1.115
p-value	0.221
q-value	0.212